# Optimization of Preparation Conditions and Storage Quality of Sour Cream Fermented by *Lactococcus lactis* grx602

**DOI:** 10.3390/foods14234159

**Published:** 2025-12-04

**Authors:** Xiaolong He, Jintao Cheng, Jianhua Tang, Chengran Guan

**Affiliations:** 1School of Tourism and Culinary Science, Yangzhou University, Yangzhou 225009, China; hxl87828361@163.com (X.H.); jhtang@yzu.edu.cn (J.T.); 2Xianghu Laboratory, Hangzhou 310027, China; jintaocheng@zju.edu.cn; 3Key Lab of Dairy Biotechnology and Safety Control, College of Food Science and Engineering, Yangzhou University, Yangzhou 225127, China

**Keywords:** sour cream, *Lactococcus lactis*, quality optimization, functional food

## Abstract

Sour cream is a functional dairy product with a unique taste and high nutritional value. In this study, *Lactococcus lactis* grx602, a strain capable of degrading milk fat, was used to produce sour cream (a fermented dairy-based emulsion, designated as Lg-sour cream). The preparation process was optimized and the quality stability during storage was evaluated. The optimal preparation parameters were identified as double-pass homogenization (15.0 MPa, 70 °C), a 1% inoculation level, and fermentation at 37 °C. During storage, Lg-sour cream exhibited acceptable fluctuations in fermentation characteristics (viable cell count, pH and acidity), continuous decreases in textural properties (hardness, adhesiveness, cohesiveness, and elasticity), progressive declines in rheological properties (water-holding capacity, apparent viscosity and viscosity), and positive variations in free fatty acid composition (types and contents of fatty acids with different chain lengths). This study provides an in-depth exploration of the nutritional value, storage stability, and processing of Lg-sour cream fermented with *L. lactis* grx602, aiming to develop a novel sour cream with potential health benefits using *L. lactis* grx602 as the starter culture.

## 1. Introduction

Functional foods are typically defined as foods or food ingredients that provide health benefits beyond their basic nutritional value [1]. Among various functional foods, sour cream is classified as a fermented dairy-based emulsion, also known as fermented cream or cultured cream. This product is manufactured through the fermentation of cream with a standardized fat content via lactic acid bacteria (LAB) [2]. After fermentation, the nutritional value of sour cream is significantly enhanced. First, proteins and lipids are broken down into more bioaccessible amino acids and free fatty acids [3]. Concurrently, this process improves the bioavailability of minerals such as calcium, phosphorus, and iron. Furthermore, it acidifies the intestinal environment to inhibit pathogenic bacteria and modulates the gut microbiota, thereby providing gastroprotective effects. Studies suggest that moderate consumption may reduce the risk of colorectal cancer and increase immunocyte activity [4]. Owing to its rich nutritional composition and creamy mouthfeel, sour cream is widely used in various culinary applications [5].

The quality of sour cream is characterized by its distinctive taste and flavor profile, glossy appearance, and homogeneous texture [6]. The mildly sour taste and buttery flavor are achieved by the activity of LAB starter cultures during fermentation [7]. The usual LAB used in sour cream production include single or mixed strains such as *Lactococcus lactis* subsp. *cremoris*, *L. lactis* subsp. *lactis*, *L. lactis* subsp. *lactis biovar*. *diacetylactis*, *Bifidobacterium*, *Lacticaseibacillus casei*, and various species and strains of *Leuconostoc* [8]. Specific strains, particularly *L. lactis* subsp. *lactis biovar*. *diacetylactis* and *Leuconostoc* species, are key contributors to flavor compound production during fermentation. In addition to starter culture, final product quality is also influenced by several other factors, including fat content, homogenization, heat treatment, fermentation conditions, and post-fermentation processes such as cooling, storage, and distribution. The key parameters for sour cream production typically include the fat content (≥18% *w*/*w*), homogenization conditions (e.g., 15 MPa at 60 °C), and fermentation parameters (typically 20–22 °C for 16–20 h) [6].

Previously, *L. lactis* grx602, which was isolated from raw milk, was shown to have beneficial properties such as acid and bile salt tolerance and cholesterol reduction. Moreover, *L. lactis* grx602 exhibited both intracellular and extracellular lipase activity [9]. When employed in sour cream production, its lipolytic capacity could contribute to flavor enhancement through the release of free fatty acids that impart aromatic complexity and potential health benefits. However, excessive lipolysis may also lead to off-flavors or textural instability if not properly controlled. Therefore, optimizing fermentation conditions is crucial for harnessing the beneficial effects while minimizing potential drawbacks. Therefore, this study aimed to optimize the fermentation process for producing sour cream with *L. lactis* grx602 and evaluate its quality stability during storage. This work will be helpful for the production of high-quality functional sour cream.

## 2. Materials and Methods

### 2.1. Media, Bacterial Isolation and Culture Conditions

*L. lactis* grx602 was previously isolated from raw milk and stored at the Jiangsu Key Laboratory of Dairy Biotechnology and Safety Control, Yangzhou University [9]. The strain was identified as *Lactococcus lactis* based on 16S rDNA sequence analysis. Subspecies-level identification was not confirmed. A patent application has been filed for this strain (ZL202110960058.0), with all patent rights residing with the corresponding author. De Man, Rogosa and Sharpe (MRS) media were used for cultivation. A pre-inoculum was prepared from a single colony and incubated in MRS broth for 24 h at 37 °C under static conditions. The activated *L. lactis* grx602 were subsequently washed twice with sterile 0.9% (*w*/*v*) sodium chloride solution before cream fermentation.

In our previous work, the initial isolation step aimed to screen for a diverse range of LAB with lipolytic activity from raw milk. MRS is a non-selective, general purpose medium formulated to support the growth of a wide spectrum of LAB, including various Lactobacillus, Leuconostoc, and Lactococcus species. This broad applicability made it suitable for our primary screening objective, which was not exclusively limited to Lactococcus at that stage. we fully acknowledge that growth on MRS alone is not diagnostic for *L. lactis*. However, to unequivocally confirm the taxonomic identity of our isolate, strain grx602, we performed multiple, repeated 16S rDNA sequencing analyses subsequent to its isolation. This molecular method provides a definitive identification, conclusively verifying the strain as *Lactococcus lactis* and thereby eliminating any potential for misidentification arising from the use of a general-purpose medium.

In summary, the combination of a broad-spectrum isolation medium (MRS) followed by rigorous molecular confirmation represents a robust and validated strategy. This approach allowed us to cast a wide net in the initial screening phase while ensuring the absolute accuracy of the final strain identification.

### 2.2. Preparation of Sour Cream Fermented by L. lactis *grx602* (Lg-Sour Cream)

The sour cream formulation consisted of the following ingredients by weight: butter (18.00%), sodium caseinate (1.10%), soybean lecithin (0.10–0.15%), skim milk powder (5.00%), monoglyceride (0.10–0.15%), carrageenan (0.02%), and tween 80 (0.05–0.10%) [9]. The butter was melted at 60 °C, and the remaining ingredients were dissolved in water. Then the melted butter and ingredients were blended under high-speed shearing (10,000 rpm) for 5 min, followed by constant stirring at 60 °C for 30 min using a magnetic stirrer to form a primary emulsion. The emulsion was homogenized and then pasteurized at 85 °C for 15 min. After cooling to approximately 30 °C, the sterilized emulsion was inoculated with 2% (*v*/*v*) of an active *L. lactis* grx602 culture and fermented until the pH reached 4.5. After fermentation, the sour cream samples were cooled and stored at 4 °C for 28 days.

To optimize the homogenization pressure and pass frequency, the emulsions were subjected to single-pass or double-pass homogenization at various pressures (5.0, 10.0, 12.5, 15.0, 17.5, and 20.0 MPa) at a constant temperature of 60 °C. The primary emulsion was homogenized using a high-pressure homogenizer (CHH-Q2000 P70, Shanghai Prime Machinery Company Limited, Shanghai, China). For the double-pass homogenization, the emulsion was passed through the homogenizer twice consecutively at the specified pressure and temperature. The two passes were performed sequentially, with an interval of 5 min between passes to maintain the processing temperature. Following homogenization, the emulsion was sterilized. With the optimal pressure established, the homogenization temperature was subsequently optimized at different temperatures (40 °C, 50 °C, 60 °C, 70 °C, 80 °C, and 90 °C). Finally, the initial inoculation ratio was optimized by fermentation at 37 °C at 1%, 2%, 3%, 5%, 8%, and 10% (*v*/*v*), followed by optimization of the fermentation temperature at 22 °C, 26 °C, 37 °C, and 42 °C with a constant initial inoculation ratio of 1% (*v*/*v*).

### 2.3. Texture Analysis

The texture of the Lg-sour cream was assessed as described previously with some modifications [10]. Eighty milliliters of Lg-sour cream samples contained in a 100 mL glass container were analyzed using a TMS-Pro texture analyzer (Food Technology Corporation, Los Angeles, CA, USA) equipped with a 12.7 mm diameter cylindrical probe. The samples were kept on ice until measurement at room temperature. The contact area was set at 5 mm, and the contact force was 0.02 N. The instrument speed was 1 mm/s. The compression distance, which was the distance of penetration from the surface of the sample, was set to 15 mm.

### 2.4. Detection of Cell Growth, Acidity, and pH

Viable bacterial counts were carried out as described previously with slight modifications [11]. Ten grams of homogenized sour cream were aseptically weighed and diluted with 90 mL of sterile saline to obtain a 10^−1^ dilution. Subsequent dilutions were prepared by transferring 1 mL of the mixture into 9 mL of diluent, followed by eight serial dilutions. The bacterial counts were determined using the pour plate method after incubation at 37 °C for 48 h at three selected optimal dilutions, and the results were expressed as the mean values. The acidity was determined by the volume of 0.10 mol/L NaOH solution required to titrate the sample to a neutral pH and the acidity was expressed as % lactic acid. pH was measured using a laboratory pH meter (Mettler Toledo FE20, Zurich, Switzerland). All trials were performed in triplicate.

### 2.5. Viscosity

Prior to measurement, Lg-sour cream was stirred 10 times clockwise and 10 times counterclockwise with a glass bar. Viscosity was measured using a viscometer (RVDV-II+, Brookfield, Engineering laboratories, Inc., Middleboro, MA, USA) with a rotor 4 and a rotation speed of 35 rpm [9].

### 2.6. Water Holding Capacity (WHC)

The WHC was detected as previously described [9]. Ten grams (m_sample = 10 g) of Lg-sour cream was centrifuged (3500 r/min, 20 min) in a pre-weighed tube (m_tube). After discarding the water, the tube was inverted for 10 min to remove residual water. The tube was then weighed (m_final), and WHC was calculated as follows:WHC (%) = ms/(mt + 10) × 100 where m_tube was the mass of the empty tube (g), m_sample was the mass of the sample, which was 10 g, and m_final was the mass of the tube plus the sediment after centrifugation and draining (g).

### 2.7. Apparent Viscosity

The apparent viscosity was determined using a Kinexus prorotational rheometer (Malvern Instruments Ltd. Malvern, UK) with modifications based on a previous study [11]. The samples were homogenized by stirring 10 times clockwise and 10 times counterclockwise before loading. Using a CP4/40 rotor, the rheology of Lg-sour cream was detected at 25 °C with a shear rate of 0.01–100 s^−1^. Data were collected every 5 s. The thixotropy (hysteresis loop area) was determined from the upward and downward flow curves.

### 2.8. Detection of Free Fatty Acid Composition

Derivatization of free fatty acids (FFAs) to fatty acid methyl esters (FAMEs) was performed according to the Chinese National Food Safety Standard GB 5009.168-2016 and the methods of Ewe and Loo [12]. Briefly, 5 mL of KOH-methanol (2 M) was added to 5.0 g of sample in a glass tube. The mixture was heated at 50 °C for 30 min, and cooled to room temperature for 2 h. Subsequently, 5 mL of isooctane was added to extract the formed FAMEs, and the mixture was vortexed vigorously. The upper organic layer containing the FAMEs was filtered through a 0.22 μm nylon membrane into a vial for GC-MS analysis.

FFA composition was detected using a GC-MS system (ISQ 7610, Thermo Fisher Scientific Inc., Waltham, MA, USA). Separation was achieved on a DB-WAX capillary column (30 m × 0.25 mm × 0.25 µm). The injector and detector were maintained at 250 °C. The oven temperature program was as follows: initial temperature at 130 °C for 5 min, increased to 220 °C at 2 °C/min and held for 5 min, and then increased to 240 °C at 2 °C/min and held for 10 min. Helium was used as the carrier gas with a flow rate of 0.8 mL/min. The injection volume was 1 μL in splitless mode, with the injector temperature set at 250 °C. The ion source (EI) temperature was 250 °C, with an ionization energy of 70 eV and a mass scan range from 45 to 450 *m*/*z*. Peaks were identified by comparing retention time with those of standard FAMEs. The relative content of each FFA was calculated as its percentage of the total area of all detected FAME peaks.

### 2.9. Statistical Analyses

The optimization experiments were conducted using a one-factor-at-a-time (OFAT) approach. This design was chosen for its effectiveness in the initial stage of process development to clearly identify the individual effect and optimal range of each key variable. All sour cream preparations and fermentation were performed as three independent biological replicates (n = 3). Measurements for each analytical parameter were conducted in triplicate. All data were analyzed by one-way ANOVA followed by Tukey’s test (SPSS 19.0) and normality (Shapiro–Wilk) and homogeneity (Levene) tests were performed before ANOVA; differences were considered significant at *p* < 0.05. Data visualization was conducted with Origin 2019.

## 3. Results and Discussion

### 3.1. Preparation Conditions of Lg-Sour Cream

#### 3.1.1. Effects of Homogenization Conditions on Lg-Sour Cream Quality

Homogenization aims to disrupt fat globules and stabilize the emulsion state through applied pressure [13]. The operational parameters for cream homogenization depend on fat content and quality requirements, with pressure, number of passes, and temperature being key determinants. Homogenization pressure modifies the fat globule size distribution and surface area in cream, thereby altering protein adsorption patterns. Both fat globule dimensions and surface protein coverage critically impact product texture and stability [14,15]. For sour cream production, current studies recommend single-pass homogenization at 10–25 MPa. The optimal pressures correlated with fat content are 15–20 MPa for 10% fat, 12–17 MPa for 18% fat, and 3–5 MPa for 38% fat [6]. In this study, as the 18% fat content of butter-based cream was used, homogenization pressures were selected to cover a broad effective range reported in the literature, with finer increments around the anticipated optimum region to accurately identify the critical pressure for optimal texture and stability. Therefore, the cream samples were processed under various pressures (5.0, 10.0, 12.5, 15.0, 17.5, and 20.0 MPa), passes (single or double), and temperatures (40, 50, 60, 70, 80, and 90 °C). After homogenization, samples were inoculated with *L. lactis* grx602. Fermentation was terminated when the pH fell below the isoelectric point of casein (pH 4.5), resulting in the formation of a gel and yielding Lg-sour cream. Homogenization effects were evaluated by detecting the texture parameters (elasticity for recoverability, cohesiveness for internal bonding, hardness for deformation resistance, adhesiveness for surface stickiness) and rheological properties (apparent viscosity).

##### Effects of Homogenization Pressure on Lg-Sour Cream Quality

With single homogenization, hardness (Figure 1A), adhesiveness (Figure 1B), cohesiveness (Figure 1C), and elasticity (Figure 1D) increased as pressure rose from 5.0 to 15.0 MPa, but decreased above 15.0 MPa. Similarly, the apparent viscosity peaked at 15.0 MPa and decreased at higher pressures (Figure 1E).

Mechanistically, increasing the homogenization pressure (from 5 to 15 MPa) reduced the fat globule size and increased the surface area, thereby facilitating casein adsorption for the formation of new interfacial membranes. These smaller globules integrated tightly with protein networks, enhancing texture and rheology. Conversely, pressures above 15 MPa led to excessive surfaces surface areas exceeding the casein coverage capacity, and introduced air bubbles, weakening structural integrity. Thus, 15.0 MPa represented the critical threshold. This nonlinear rheological response aligned with dairy processing phenomena [16,17].

Correlation analysis was conducted to elucidate the interrelationships between the process parameters and the quality attributes. The results revealed a nonsignificant relationship between homogenization pressure and cohesiveness (*p* > 0.05). However, pressure showed significant positive correlations with hardness, adhesiveness, and apparent viscosity (*p* < 0.05), and a very strong positive correlation with elasticity (*p* < 0.01). Furthermore, elasticity itself was strongly correlated with other parameters (*p* < 0.01). These data indicated that pressure predominantly modulated elasticity, hardness, adhesiveness, and viscosity. In particular, elasticity was a primary and sensitive indicator of microstructural changes induced by homogenization. In addition, hardness was positively correlated with cohesiveness (*p* < 0.05) and significantly correlated with adhesiveness, elasticity, and apparent viscosity (*p* < 0.01), aligning with the mechanistic analysis above. This pattern of correlations indicated that increasing homogenization pressure primarily enhanced the elasticity and interconnectedness of the protein-fat network, which in turn drove improvements in overall firmness (hardness) and flow resistance (viscosity). This mechanistic understanding revealed that 15.0 MPa was the optimal pressure, as it maximized the strength of this cohesive network before the structural breakdown observed at higher pressures.

##### Effects of Double-Pass Homogenization on Lg-Sour Cream Quality

In dairy processing, the usual homogenization methods include single, two-stage, and double-pass homogenization. Specifically, two-stage homogenization involves sequential passage through two homogenization valves within the same homogenizer, whereas double-pass homogenization refers to two consecutive homogenization cycles in one homogenizer or one cycle each in two separate homogenizers [6]. In this experiment, butter-based cream was homogenized twice at the same pressure before inoculation with *L. lactis* grx602 for fermentation. Compared with single-pass homogenization, double-pass homogenization at the same pressure significantly improved all texture and rheological parameters (Figure 1A–F). The pressure during double processing was positively correlated with hardness (*p* < 0.05) but not with other parameters. Hardness showed strong positive correlations with other parameters (*p* < 0.01). Compared to single-pass homogenization, double-pass homogenization further increased sour cream hardness.

From a physicochemical perspective, the primary advantage of double homogenization lies in the secondary optimization of fat globule size distribution. Fat globule fragments formed during the first homogenization undergo refinement during the second homogenization, promoting a more uniform distribution of casein coating the fat globule membranes [18,19]. The significant positive correlation between homogenization pressure and hardness (*p* < 0.05) can be mechanistically explained by the reduction in fat globule size and the formation of a more robust casein-coated fat globule network, which collectively increases the resistance of the gel to deformation [17]. This microstructural modification directly increased bacterial interfacial contact, accelerating the acid production rate and leading to a gel network with a more uniform, compact texture and greater resistance to deformation (manifested as increased hardness), thereby simultaneously improving both textural and rheological properties.

To prevent fat globule membrane aggregation and maintain product volume, two-stage homogenization is not recommended for creams with fat contents exceeding 10–15%. However, double-pass homogenization is sometimes recommended for producing sour cream with higher hardness. For example, Emmons and Tuckey [20] recommended double-pass homogenization at 17.2 MPa for creams with fat contents of 18.6% and 10.5–12% fat at 74 °C and 43 °C, respectively. In this study, Lg-sour cream subjected to double-pass homogenization presented textural and rheological properties superior to those of Lg-sour cream subjected to single homogenization.

##### Effects of Homogenization Temperature on Lg-Sour Cream Quality

The typical homogenization temperatures for cream range from 40 °C to 85 °C [6]. In this work, butter-based cream was homogenized at 40 °C, 50 °C, 60 °C, 70 °C, 80 °C, and 90 °C. Elasticity (Figure 2A), hardness (Figure 2C), and adhesiveness (Figure 2D) initially increased but then decreased with increasing temperature, peaking at 70 °C, whereas cohesiveness (Figure 2B) first decreased and then increased. Cohesiveness provided complementary information about the gel’s internal integrity and resilience post-fracture, which did not always directly correlate with its initial resistance to deformation (hardness). The distinct behavior of cohesiveness might be attributed to its specific sensitivity to the recovery of the protein-fat network. At lower homogenization temperatures (40–60 °C), the formation of a stronger, more cross-linked gel (as indicated by increasing hardness) may have resulted in a structure that, once fractured during the first compression, did not recover as effectively for the second compression, leading to a lower cohesiveness value. As the temperature increased to the optimal 70 °C, a more uniform and balanced network was formed, which not only was strong but also exhibited better structural integrity and recovery after the initial deformation, thereby increasing the cohesiveness ratio. This phenomenon highlighted that cohesiveness measured a different aspect of texture—internal bonding and resilience—compared to the primary strength parameters like hardness and elasticity. Ahmed et al. [21] also found that homogenization at higher temperatures yielded sour cream with superior texture. Furthermore, Lg-sour cream exhibited characteristics of a pseudoplastic fluid at all temperatures (Figure 2E). At the same shear rates, homogenization temperature significantly influenced the fermented gel network by affecting casein denaturation and the physical state of fat globules [22].

In this work, within 40–70 °C, higher temperatures improved butter dispersion and homogenization efficiency, leading to a more compact and robust gel network with improved textural and rheological properties after fermentation. Above 70 °C, especially near 90 °C, the texture and rheology of the Lg-sour cream decreased. Thus, 70 °C was identified as the optimal homogenization temperature. Homogenization near this temperature maximized the beneficial effects of protein denaturation (reinforcement) and the micronization of solid fat while simultaneously avoiding the detrimental effects of protein aggregation and liquid fat interference induced by excessive heating.

#### 3.1.2. Effects of Inoculum Size on Lg-Sour Cream Quality

The inoculum size directly influenced the acidification rate and fermentation duration, thereby affecting the gelation kinetics of sour cream [23]. Here, *L. lactis* grx602 with different inoculum sizes were inoculated into butter-based cream with double-pass homogenization at 70 °C. With the increase in inoculation amount, the elasticity (Figure 3A), hardness (Figure 3C), and adhesiveness (Figure 3D) of Lg-sour cream decreased, whereas its cohesiveness (Figure 3B) increased. Concurrently, the apparent viscosity decreased at equivalent shear rates as the inoculum size increased (Figure 3E). The optimal rheological and textural properties were observed at a 1% inoculation.

Along with *L. lactis* grx602 growth, acids were produced, leading to a gradual decline in pH. As the pH decreased, casein aggregated at the isoelectric point, progressively forming a uniform and dense three-dimensional gel network in the Lg-sour cream. This slow process of aggregation and rearrangement was crucial for the formation of a high-strength, fine-textured gel structure. A high inoculation size would accelerated bacterial growth and acid production rates, causing instantaneous and disordered protein aggregation. This resulted in the formation of a coarse, loose, and heterogeneous weak network structure, ultimately compromising key textural properties. Correlation analysis confirmed significant negative correlations between inoculum size and all texture and rheological parameters (*p* < 0.01), indicating that increasing the inoculation level of *L. lactis* grx602 adversely affected the texture and rheological properties of Lg-sour cream, thereby diminishing its quality. These results also confirmed that faster acidification at higher inoculation levels universally and detrimentally affected the gel network formation, thereby strengthening the rationale for selecting the lowest inoculation level (1%) that achieved the target pH within a reasonable time.

#### 3.1.3. Effects of Fermentation Temperature on Lg-Sour Cream Quality

Fermentation temperature modulated starter growth, consequently affecting product quality [24]. Samples prepared under optimal homogenization and inoculation conditions were fermented at 22 °C, 26 °C, 37 °C, and 42 °C. Lg-sour cream produced at 37 °C presented superior elasticity (Figure 4A), cohesiveness (Figure 4B), hardness (Figure 4C), adhesiveness (Figure 4D), and apparent viscosity (Figure 4E).

Fermentation temperature showed highly significant positive correlations (*p* < 0.01) with hardness, cohesiveness, adhesiveness, and apparent viscosity, and a significant positive correlation (*p* < 0.05) with elasticity. These data indicated that fermentation temperature significantly affected the texture and rheological properties of Lg-sour cream.

### 3.2. Lg-Sour Cream Quality During Storage

Lg-sour cream prepared under the optimal conditions (double-pass homogenization at 15.0 MPa and 70 °C, 1% inoculation level, and fermentation at 37 °C with *L. lactis* grx602) was stored at 4 °C, and its quality stability was evaluated over a 28-day period. Specially, stability in this study does not imply that all properties remain unchanged, but rather that the product remains within acceptable quality limits until the end of its shelf life.

#### 3.2.1. Fermentation Characteristics

The viable cell count of *L. lactis* grx602 peaked at 11.51 log CFU/mg upon fermentation completion (day 1) and subsequently declined to 9.43 log CFU/mg on day 14, likely due to inhibition by the low storage temperature (Figure 5A). Nevertheless, the count remained above 7.00 log CFU/mg throughout the entire storage period. This level significantly surpasses the requirement of national standards and is also higher than the LAB count reported in commercial butter by Dagdemir et al. [25].

Changes in pH and acidity followed a trend consistent with the viable cell count (Figure 5B). The pH decreased from 4.51 to 4.33, and the acidity increased from 0.64% lactic acid to 0.67% lactic acid during the first 14 days, stabilizing thereafter. By day 28 of storage, the pH and acidity of the Lg-sour cream were 4.32 and 0.68% lactic acid, respectively. This finding aligned with the growth trend of *L. lactis* grx602, wherein LAB-produced organic acids caused a decrease in pH and an increase in the titratable acidity [26]. This finding aligned with existing research findings [27,28]. In addition to the organic acidosis, free fatty acids (FFAs) generated through the lipase-mediated hydrolysis of milk fat further augment acidity [8]. Given that the generally accepted pH range for fermented dairy products is 4.20 to 4.80, and considering the USDA requirement for sour cream acidity (minimum 55.56 °T), the Lg-sour cream met the quality standards for sour cream throughout the entire storage period [8].

#### 3.2.2. Textural Properties

With prolonged storage time, Lg-sour cream exhibited continuous decreases in hardness (Figure 6A), adhesiveness (Figure 6B), cohesiveness (Figure 6C), and elasticity (Figure 6D). These results were consistent with those of previous studies [29,30]. The observed textural changes correlated with microbial growth dynamics. During the first 14 days of storage, although the *L. lactis* grx602 cells continuously produced metabolites such as organic acids and proteases, the water-holding capacity (WHC) of the casein micelles remained temporarily stable. This manifested as negligible declines in hardness and elasticity. However, the reduction in interfacial tension caused by organic acids and other metabolites led to significant decreases in adhesiveness and cohesiveness. After the initial 14 days of storage, the metabolism of *L. lactis* grx602 slowed, further accumulation of metabolites subsequently resulting in pronounced reductions in hardness and adhesiveness.

#### 3.2.3. Rheological Properties

WHC, viscosity, and apparent viscosity are commonly used to characterize the rheological properties of dairy products. The WHC reflected the ability of the casein gel network to entrap water and directly determines the extent of whey separation during storage [31]. Viscosity governed the rate of fat separation by reflecting the strength of interactions between fat globules and the protein matrix [32]. Apparent viscosity, which was indicative of the thixotropy of non-Newtonian fluids, inhibited gel structure collapse through shear-recovery resistance [33]. Collectively, these three parameters reflected the stability of dairy product texture. During storage, both the WHC (Figure 7A) and apparent viscosity (Figure 7C) of the Lg-sour cream decreased progressively. Viscosity remained relatively stable for the first 14 days but declined significantly thereafter (Figure 7B). After 28 days of storage, the WHC, viscosity, and apparent viscosity decreased by 27.82%, 16.78%, and 62.14%, respectively. The decline in WHC, viscosity, and apparent viscosity during storage were mainly caused by *L. lactis* grx602 metabolites, especially organic acids and enzymes. Metabolic acid production caused disintegration of the casein micelle structure. These exposed water-binding groups, then bind free hydrogen ions, consequently reducing the WHC [34]. The progressively acidic environment induced protein denaturation. Concurrently, lipase production by *L. lactis* grx602 hydrolyzed fat. These processes collectively disrupted the fat globule membrane structure, leading to reductions in the sour cream’s WHC, viscosity, and apparent viscosity.

#### 3.2.4. Free Fatty Acid Composition

The quantitative variation in fatty acid content directly correlated with the flavor and nutritional quality of sour cream. As shown in Table 1, 17, 20 and 22 distinct free fatty acids (FFAs) were detected in the Lg-sour cream at day 1, day 7 and day 7 later, respectively. Of these, hexanoic acid (C6:0), undecanoic acid (C11:0) and eicosatrienoic acid (C20:3n6) were detected at day 7, while hendecanoic acid (C11:0) and tridecanoic acid (C13:0) emerged at day 7 later. These FFAs were predominantly categorized as short-chain fatty acids (SCFAs), medium-chain fatty acids (MCFAs), long-chain fatty acids (LCFAs), and very-long-chain fatty acids (VLCFAs), with LCFAs constituting the highest proportion. Among the persistently dominant fatty acids throughout storage, LCFAs of myristic acid (C14:0), palmitic acid (C16:0), stearic acid (C18:0), and oleic acid (C18:1n9c) exhibited the highest concentrations, with palmitic acid (C16:0) demonstrating particular abundance. These findings aligned with the FFAs composition of sour cream separately fermented with *Bifidobacterium lactis*, *L. acidophilus* and *L. rhamnosus* [35].

The FFA content generally exhibited an initial increase followed by either a gradual decline or stabilization. The observed variation in certain fatty acid concentrations might reflect both bacterial assimilation and their bioconversion into volatile flavor compounds [36,37]. SCFAs and MCFAs, including hexanoic acid(C6:0), acid octanoic acid (C8:0), acid decanoic acid (C10:0), and lauric acid (C12:0), exhibited low olfactory thresholds and high volatility. These compounds imparted distinctive lipid-derived aromas to cream, significantly contributing to the flavor profile of sour cream [38]. During storage stages, the increased concentrations of hexanoic acid (C6:0), octanoic acid (C8:0), and decanoic acid (C10:0) could enhance the aromatic quality of Lg-sour cream.

Saturated free fatty acids (SFFAs), characterized by saturated bonds, served as essential energy substrates for human metabolic processes. The predominant SFFAs in Lg-sour cream were myristic (C14:0), palmitic (C16:0), and stearic acid (C18:0) which progressively accumulated throughout storage and reached peak concentrations of 13.43%, 31.01%, and 12.71%, respectively. Notably, unsaturated free fatty acids (UFFAs) were detected in Lg-sour cream, including both monounsaturated free fatty acids (MUFFAs) and polyunsaturated free fatty acids (PUFFAs). Oleic acid (C18:1), the predominant MUFFAs, reached a maximum concentration of 18.34% during storage. This compound had been shown to exert various potential physiological properties, including hypocholesterolemic, hypoglycemic, and lipid-regulatory effects, conferring preventive benefits against cardiovascular diseases [39]. PUFFAs, being essential fatty acids requiring dietary intake, were dominated by linoleic acid (C18:2n6) in Lg-sour cream. Additionally, conjugated linoleic acid (C18:2c9t11), α-linolenic acid (C18:3n3), arachidonic acid (C20:4n6), and eicosapentaenoic acid (EPA, C20:5n3) were identified among the PUFFAs. The increase in PUFFAs and MUFFAs content suggested potential nutritional relevance; further biological studies would be required to confirm any health effects.

## 4. Conclusions

This study optimized the preparation of Lg-sour cream using *L*. *lactis* grx602, establishing the following optimal conditions: double-pass homogenization at 15.0 MPa and 70 °C, 1% inoculation, and fermentation at 37 °C. Under these conditions, Lg-sour cream exhibited desirable fermentation characteristics, textural and rheological properties, and a diverse FFA profiles. During 28 days of storage, the viable cell counts remained above 7.00 log CFU/mg, and the pH and acidity remained within acceptable ranges. Although the textural and rheological properties gradually decreased, the Lg-sour cream maintained microbiological safety, metabolic activity, and compositional stability, thereby sustaining its overall acceptability and functional potential throughout the 28-day shelf life. This study highlighted that *L. lactis* grx602 not only served as a functional starter culture but also contributed to product stability by maintaining high viability. The progressive increase in free fatty acids, particularly flavor-active short- and medium-chain fatty acids, suggested that the product may develop a more nutritional and desirable flavor profile over time, which is a positive attribute for the industry. Moreover, the presence of various nutritionally beneficial unsaturated FFAs highlighted the potential health value of Lg-sour cream. However, these potential physiological benefits, inferred from the compositional data, should be further verified and quantified through future in vivo model studies to confirm their bioavailability and actual health impacts. These findings supported the use of *L. lactis* grx602 for the production of high-quality functional sour cream with improved nutritional and sensory attributes.

## Figures and Tables

**Figure 1 foods-14-04159-f001:**
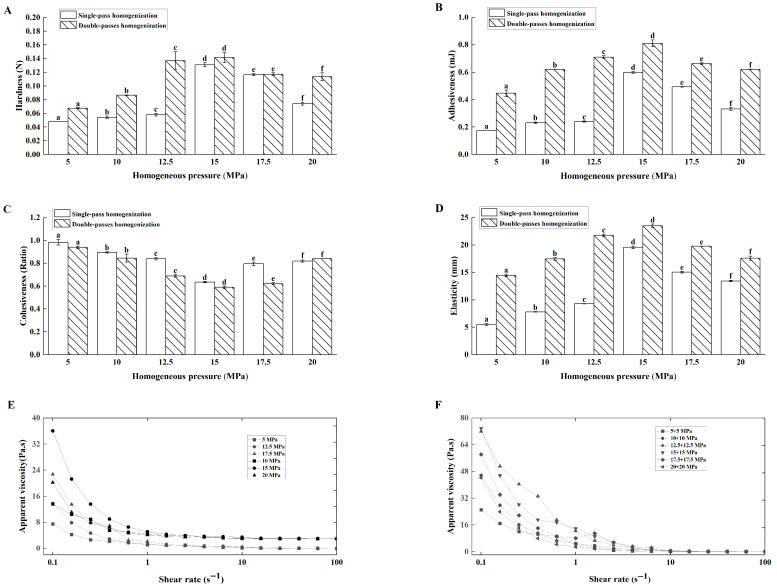
Effects of homogenization pressure and time on the textural properties and apparent viscosity of the Lg-sour cream. (**A**) Hardness, (**B**) Adhesiveness, (**C**) Cohesiveness, (**D**) Elasticity, (**E**) Apparent viscosity after single-pass homogenization, and (**F**) Apparent viscosity after double-pass homogenization. Different lowercase letters indicated significant differences (*p* < 0.05).

**Figure 2 foods-14-04159-f002:**
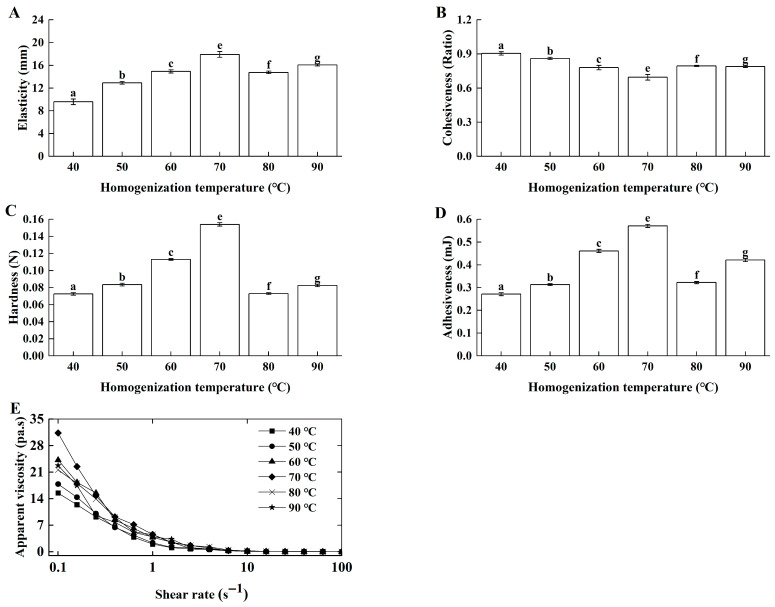
Effects of homogenization temperature on the textural properties and apparent viscosity of the Lg-sour cream. All the samples were homogenized at 15.0 MPa with double passes. (**A**) Elasticity, (**B**) Cohesiveness, (**C**) Hardness, (**D**) Adhesiveness and (**E**) Apparent viscosity. Different lowercase letters indicated significant differences (*p* < 0.05).

**Figure 3 foods-14-04159-f003:**
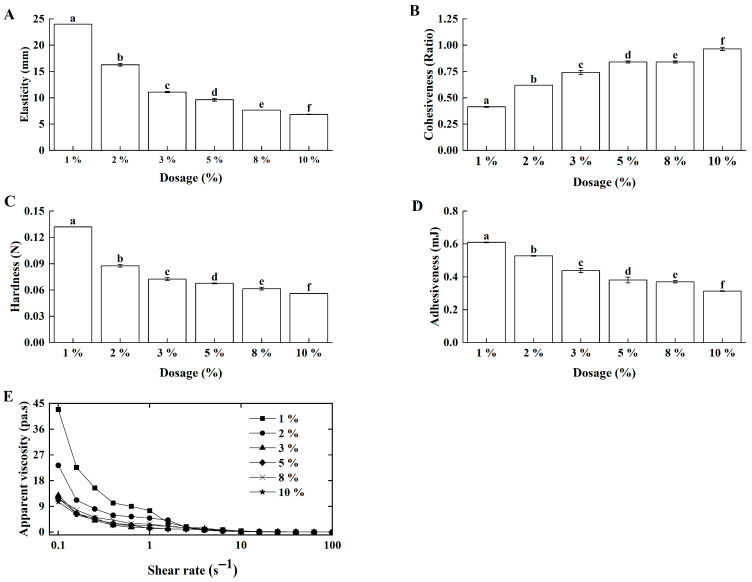
Effects of inoculation size on the textural properties and apparent viscosity of the Lg-sour cream. All the samples were homogenized at 15.0 MPa with double passes at 70 °C. (**A**) Elasticity, (**B**) Cohesiveness, (**C**) Hardness, (**D**) Adhesiveness, and (**E**) Apparent viscosity. Different lowercase letters indicated significant differences (*p* < 0.05).

**Figure 4 foods-14-04159-f004:**
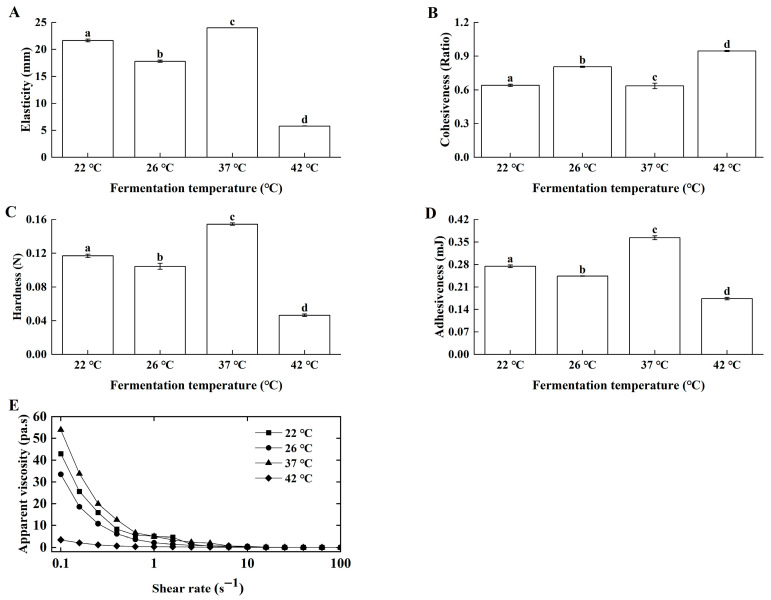
Effects of fermentation temperature on the textural properties and apparent viscosity of the Lg-sour cream. All the samples were homogenized at 15.0 MPa with double passes at 70 °C, and then with 1% inoculum size (*v*/*v*), after which fermentation was performed at different temperatures. (**A**) Elasticity, (**B**) Cohesiveness, (**C**) Hardness, (**D**) Adhesiveness, and (**E**) Apparent viscosity. Different lowercase letters indicated significant differences (*p* < 0.05).

**Figure 5 foods-14-04159-f005:**
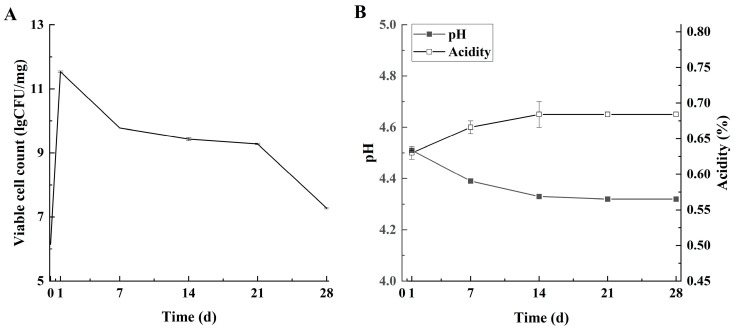
The fermentation characteristics of the Lg-sour cream during storage at 4 °C for 28 d. (**A**) Viable cell count, (**B**) pH and acidity. Viable counts were expressed as CFU/g. Values represent mean ± SD of replicate plates (n = 3).

**Figure 6 foods-14-04159-f006:**
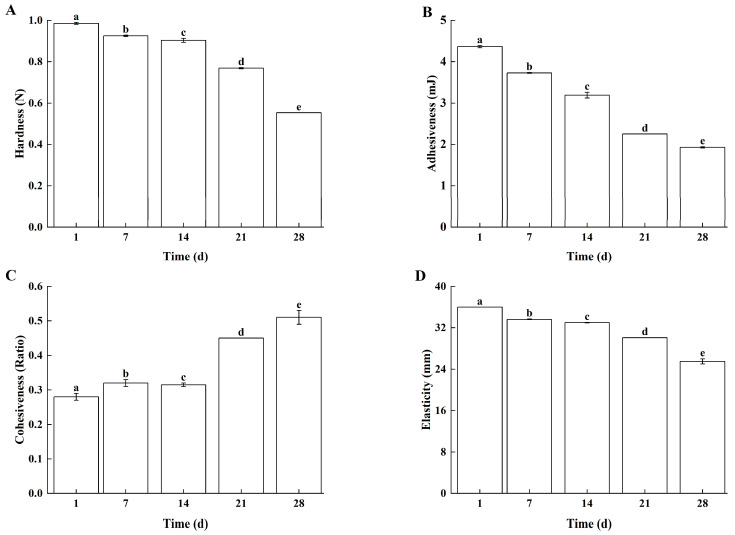
The textural properties of Lg-sour cream during storage at 4 °C for 28 d. (**A**) Hardness, (**B**) Adhesiveness, (**C**) Cohesiveness, and (**D**) Elasticity. Different lowercase letters indicated significant differences (*p* < 0.05).

**Figure 7 foods-14-04159-f007:**
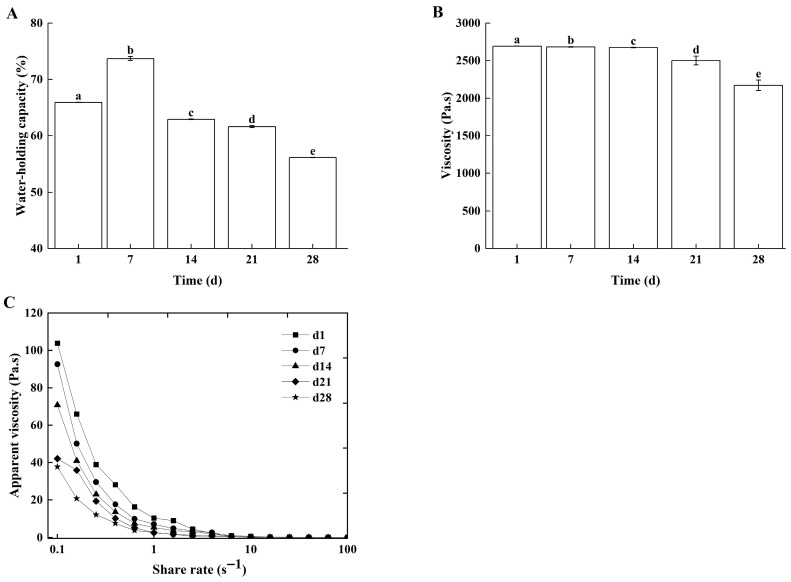
The rheological properties of Lg-sour cream during storage at 4 °C for 28 d. (**A**) WHC, (**B**) viscosity, and (**C**) Apparent viscosity. Different lowercase letters indicated significant differences (*p* < 0.05).

**Table 1 foods-14-04159-t001:** Relative composition of free fatty acids (% of total FFAs) in Lg-sour cream during storage.

FFAs		Time				
		1 d	7 d	14 d	21 d	28 d
SCFA	C6:0	/	0.03 ± 0.01 ^d^	0.30 ± 0.04 ^a^	0.19 ± 0.03 ^c^	0.23 ± 0.03 ^b^
MCFA	C8:0	/	0.14 ± 0.02 ^d^	0.47 ± 0.05 ^a^	0.41 ± 0.04 ^b^	0.38 ± 0.03 ^c^
	C10:0	0.31 ± 0.03 ^e^	1.25 ± 0.07 ^d^	2.11 ± 0.11 ^a^	2.03 ± 0.10 ^b^	1.85 ± 0.09 ^c^
	C11:0	/	/	0.14 ± 0.02 ^a^	0.13 ± 0.02 ^a^	0.13 ± 0.02 ^a^
	C12:0	1.92 ± 0.05 ^e^	3.23 ± 0.09 ^d^	4.14 ± 0.13 ^a^	4.03 ± 0.12 ^b^	3.82 ± 0.11 ^c^
	C13:0	/	/	0.12 ± 0.02 ^a^	0.11 ± 0.02 ^a^	0.11 ± 0.02 ^a^
	C14:0	10.1 ± 0.15 ^e^	12 ± 0.18 ^d^	13.43 ± 0.21 ^a^	13.12 ± 0.20 ^b^	12.98 ± 0.19 ^c^
	C14:1	0.54 ± 0.03 ^b^	1.03 ± 0.05 ^a^	1.09 ± 0.05 ^a^	1.07 ± 0.05 ^a^	1.06 ± 0.05 ^a^
	C15:0	1.21 ± 0.05 ^c^	1.69 ± 0.05 ^ab^	1.73 ± 0.05 ^a^	1.62 ± 0.04 ^b^	1.71 ± 0.05 ^a^
LCFA	C16:0	31.01 ± 0.25 ^a^	27.75 ± 0.23 ^e^	29.36 ± 0.24 ^c^	30.11 ± 0.24 ^b^	28.28 ± 0.23 ^d^
	C16:1	1.39 ± 0.05 ^b^	1.84 ± 0.06 ^a^	1.77 ± 0.06 ^a^	1.78 ± 0.06 ^a^	1.73 ± 0.06 ^a^
	C17:0	0.56 ± 0.03 ^c^	0.78 ± 0.04 ^a^	0.7 ± 0.04 ^ab^	0.68 ± 0.04 ^b^	0.73 ± 0.04 ^ab^
	C18:0	12.45 ± 0.15 ^a^	12.71 ± 0.15 ^a^	11.5 ± 0.14 ^b^	11.56 ± 0.14 ^b^	12.55 ± 0.15 ^a^
	C18:1n9c	18.34 ± 0.20 ^a^	16.35 ± 0.19 ^b^	13.49 ± 0.17 ^d^	15.13 ± 0.18 ^c^	15.42 ± 0.18 ^c^
	C18:2n6c	1.89 ± 0.05 ^c^	2.22 ± 0.05 ^a^	2.17 ± 0.05 ^ab^	2.27 ± 0.05 ^a^	2.12 ± 0.05 ^b^
	C18:3n3	0.71 ± 0.03 ^c^	0.96 ± 0.03 ^a^	0.85 ± 0.03 ^b^	0.84 ± 0.03 ^b^	0.87 ± 0.03 ^b^
	C18:2c9t11	1.52 ± 0.04 ^c^	2.03 ± 0.05 ^a^	1.67 ± 0.04 ^b^	1.69 ± 0.04 ^b^	1.75 ± 0.04 ^b^
VLCFA	C20:0	0.19 ± 0.01 ^a^	0.21 ± 0.01 ^a^	0.20 ± 0.01 ^a^	0.19 ± 0.01 ^a^	0.22 ± 0.01 ^a^
	C20:1n9	0.15 ± 0.01 ^b^	0.18 ± 0.01 ^a^	0.11 ± 0.01 ^c^	0.11 ± 0.01 ^c^	0.12 ± 0.01 ^c^
	C20:3n6	/	0.11 ± 0.01 ^a^	0.09 ± 0.01 ^a^	0.09 ± 0.01 ^a^	0.09 ± 0.01 ^a^
	C20:4n6	0.15 ± 0.01 ^c^	0.25 ± 0.02 ^a^	0.14 ± 0.01 ^c^	0.20 ± 0.02 ^b^	0.15 ± 0.01 ^c^
	C20:5n3	0.08 ± 0.01 ^c^	0.20 ± 0.01 ^a^	0.14 ± 0.01 ^b^	0.14 ± 0.01 ^b^	0.12 ± 0.01 ^b^

Values represent mean ± SD (n = 3). Means in the same row with different superscript letters indicated significantly different (*p* < 0.05).

## Data Availability

The original contributions presented in the study are included in the article, further inquiries can be directed to the corresponding author.

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
