# Peer review of "Optimization of Preparation Conditions and Storage Quality of Sour Cream Fermented by Lactococcus lactis grx602"

_foods, 2025, doi:10.3390/foods14234159_

Round 1
Reviewer 1 Report
Comments and Suggestions for Authors
Please find the attachment.

Author Response
- “L53 L. lactis grx602, capable of degrading butter
I can only find a patent for this claim. Could authors provide reference of previously published paper. For lipolytic activity of L. lactis grx602 there is no up to date study. Moreover, Lactococcus lactis ssp. lactis is one of the most important starter bacteria used in dairy technology, and various studies have confirmed that it lacks lipolytic activity. Many LAB show limited and weak lipase activity. However, it is interesting that sometimes methodology could give false negative results. Strains that had no lipolytic activity in agar could exert lipolytic activity in dairy products. Could the authors discuss this? Also, in the introduction, outline the advantages and disadvantages of adding a strain capable of degrading milk fat.”
Respond:Thank you for your comment. The strain L. lactis grx602 was indeed previously isolated and characterized in our laboratory, and its lipolytic activity was initially reported in a Chinese patent. The screening and isolation process of this strain, the enzymatic properties of its lipase, and the probiotic characteristics and applications of the strain are detailed in a full-length manuscript under submission and are not presented in this article.
Regarding the lipolytic activity of L. lactis grx602, we agree with the reviewer that many Lactococcus strains are considered to have weak or undetectable lipase activity under standard agar-based assays. However, as the reviewer rightly pointed out, the expression of lipolytic activity can be highly dependent on the assay conditions and the growth medium. In our another submitted article, both intracellular and extracellular lipase activities in L. lactis grx602 were detected which was lower than that of lipase found in bacteria like Bacillus and fungi [1]. Here is the method for strain isolation and detection of lipase activity.
Isolation and identification of L. lactis grx602 : Firstly, by enrichment on the MRS solid medium, 66 colonies which were morphological like the lactic acid bacteria were obtained from raw milk and numbered sequentially 1 to 66. Then, only 24 colonies could grow on the MRS-1 agar plates (glucose in MRS replaced by butter). Further, to eliminate the influence of the organic carbon and nitrogen source (yeast nitrogen, tryptone, and beaf extract), only 10 strains were chosen which could grow on the MRS-2 medium (removing yeast nitrogen, tryptone, and beaf extract in MRS-1) but could not grow on the MRS-3 medium (removing butter in MRS-2) (Fig. 1A). Finally, strain 23, showing the best-growth, was selected and further demonstrated to be positive by Gam staining analysis (Fig. 1B). The 16S rDNA of strain 23 was amplified and sequenced (Fig. 1C). By blasting with the deposited sequences in GenBank, strain 23 possessed a 16S rDNA sequence with 99% similarity to Lactococcus lactis sysI9 (GeneBank: OL823012.1) (Fig. 1 D). Combining with the morphological properties, strain 23 was renamed as Lactococcus lactis grx602 (GenBank: PQ554972) and then stored in the China General Microbiological Culture Collection Center (CGMCC No. 22692).
Lipase activities assay: Modified medium (g/L) used for lipase production was composed of butter (20.0), glucose (5.0), tryptone (10), sodium acetate (5.0), magnesium sulfate heptahydrate (0.05), dipotassium hydrogen phosphate (2.0), diammonium hydrogen citrate (2.0), manganous sulfate (0.05). L. lactis grx602 was incubated in the modified medium at 37 ℃ for 72 h and sampled every 12 h. After cultivation, cells and supernatant were separated by centrifugation. The supernatant was directly used for extracellular lipase detection and fatty acid composition. The harvested cells were washed by Tris-HCl buffer (50 mM, pH 7.0) for three times, suspended in an appropriate diluted ratio, disrupted on ice by a cell disruptor (Constant System Ltd., England), and centrifugated (10 min, 4 ℃, 10000 r/min). The resulting supernatant of the disrupted cells was used for intracellular lipase detection.
Using p-nitrophenyl palmitate (pNPP, Sigma, USA) as the substrate, the lipase activity was spectrophotometrically determined as previously described [2, 3]. Briefly, a 25 mM p-NPP stock solution was mixed with Tris-HCl buffer (50 mM, pH 8.0) in a volume ratio 1:9, and then 0.1% gum arabic was added, yielding solution A. For the reaction, 50 μL of solution A and 50 μL of lipase solution were mixed and incubated at 37 ℃ for 15 min. The reaction was stopped by adding 400 μL of trichloroacetic acid (0.5 M), and then 400 μL of sodium carbonate (0.5 M) was added for color development. After centrifugation (12000×g, 1min), liberated p-nitrophenol was measured at 410 nm in a Multiskan GO microplate reader (Thermo Fisher Scientific, USA). Control reactions were prepared by adding trichloroacetic acid to the lipase sample before substrate addition. One unit (U) of lipase activity is defined as the amount of enzyme liberating 1 μmol of p-nitrophenol per minute under standard assay conditions. Enzyme assays were performed in triplicate.
We agree with the reviewer that some LAB strains exhibit lipolytic activity only in complex dairy matrices due to substrate induction, co-factor availability, or post-translational modifications. We have now added a discussion on this point in the revised manuscript (Lines 58–62) as follows: “L. lactis grx602 exhibited both intracellular and extracellular lipase activity. When employed in sour cream production, its lipolytic capacity could contribute to flavor enhancement through the release of free fatty acids that impart aromatic complexity and potential health benefits. However, excessive lipolysis may also lead to off-flavors or textural instability if not properly controlled. Therefore, optimizing fermentation conditions is crucial to harness the beneficial effects while minimizing potential drawbacks.”
- “L63 isolated from our strain collection.
What collection would that be? How did that strain end up in your collection? Is it commercially available, is it an isolate, where can we see its characteristics?
Question regarding product preparation: Is L. lactis grx602 only bacteria that is used, or there are other starter bacteria?”
Respond:The isolation, screening, and identification procedures for L. lactis grx602 have been detailed in a separate publication [1], which was previously unpublished at the time of this submission but is now publicly available. L. lactis grx602 is not a commercial strain and is available from the corresponding author for academic research purposes. In the present study, this strain served as the sole microbial culture for sour cream fermentation.
- “L75 After cooling to approximately 30 ℃, 75 L. lactis grx602 was inoculated for fermentation until the pH reached 4.5.In what amount (CFU/mL). Specify the number of bacteria added. Figure 5. X assix lgcfu/ml replace with log CFU/mL.”
Respond:The inoculation was indeed performed using a 2% (vol/vol) ratio of the bacterial culture. To ensure the reproducibility of our experiment, we have now supplemented the manuscript with the specific bacterial cell count at the time of inoculation. The text has been revised as follows: After cooling to approximately 30 ℃, the sterilized emulsion was inoculated with 2% (vol/vol) of an active L. lactis grx602 culture and fermented until the pH reached 4.5.
Figure 5. X assix lgcfu/ml has been replace with log CFU/mL.
- “L374-378 Bifidobacterium lactis, L. acidophilus and L. rhamnosus[35]. Italic”
Respond:Done.
- “General comment: https://patents.google.com/patent/CN113736694A/en?q=A23C15%2f165
Yangzhou University is the patent holder, as the authors are from the same university, if this is their patent they should state that.”
Respond:Thank you for you kind suggestion. The patent of this strain has been stated in the revised manuscript (2.1 Media, bacterial isolation and culture conditions)_as follows: A patent application has been filed for this strain (ZL202110960058.0), with all patent rights residing with the corresponding author.
Reference:
- Peng Y., Chengran G., Yuan Y., Yiping L., Xin W., Renqin Y., Chenchen Z., Dawei C., Yujun H., Ruixia G.. Characterization of Lactococcus lactisgrx602: A novel lipase-producing probiotic strain for nutritional and sensory enhancement of sour cream. Food Bioscience, 2025,107918.org/10.1016/j.fbio.2025.107918.
Reviewer 2 Report
Comments and Suggestions for Authors
The manuscript "Optimization of preparation conditions and storage quality of fermented sour cream by Lactococcus lactis grx602" presents an investigation into the optimization of the preparation process and storage stability of sour cream fermented with Lactococcus lactis grx602. The topic is relevant and aligns with the current interest in functional dairy products and microbial optimization. The study is well structured. The authors provide a thorough analysis of processing parameters and quality evaluation during storage.
However, the manuscript would benefit from clarifications, improved English expression, and stronger statistical and methodological rigor. Some sections also require better justification and interpretation of results.
The novelty of using L. lactis grx602 should be stated more clearly in my opinion. Is this strain significantly different in function or performance compared to other L. lactis strains already applied in dairy fermentation? The introduction should include a comparative context with previous optimization studies or strain applications in similar products. Clarify the novelty and unique contribution of L. lactis grx602.
Experiments are properly executed but lack advanced statistical design (e.g., DOE/RSM). Replication details are insufficient, and correlation analysis requires stronger interpretation. Why did the authors choose a single-variable approach? The ability to test for effects can be disabled. It is unclear whether the experiments were biological or visual replicates. Experimental replications and repetitions should be used for each test. Justify the experimental design and replication strategy.
Although ANOVA and Tukey’s test are mentioned, p-values and confidence intervals are not consistently reported. Statistical significance should be indicated on all figures and tables. Correlation analyses are presented without sufficient explanation of how they contribute to optimization—please clarify their interpretation and purpose.
The storage evaluation section is comprehensive but overly descriptive. It should emphasize key findings and practical implications (e.g., shelf life recommendations, industrial relevance). The authors claim that Lg-sour cream “maintained overall quality and stability” during storage. However, textural and rheological parameters decreased significantly. Please justify how quality stability was defined.
Figures are informative but overly complex. Consider simplifying axis labels and ensuring that font sizes are legible. Statistical letters (a, b, c) should be placed consistently across subfigures. Table 1 lacks standard deviations and should include statistical analysis of FFA changes over time. Strengthen statistical reporting and visual presentation.
The manuscript requires language editing by a native or professional English editor. Common issues include: incorrect pluralization and article use, overuse of passive constructions, and inconsistent verb tenses. Example corrections:
- “This study provided an in-depth exploration...” → “This study provides an in-depth exploration...”
- “The optimal preparation parameters were determined as…” → “The optimal preparation parameters were identified as…”. Perform professional English editing prior to resubmission.
Make corrections:
- Please define all abbreviations (e.g., WHC, FFA, LCFA) at their first appearance in both the abstract and main text,
- provide the manufacturer details (city, country) for all instruments and materials mentioned,
- in Section 2.2, check the formulation units (“18.00% butter”) — confirm whether this refers to total fat content or ingredient proportion,
- some references are outdated or lack DOIs. Ensure consistency with MDPI citation style,
The study has scientific merit and potential value for the dairy and functional food research community. However, it requires substantial revision to improve methodological clarity, statistical robustness, and English readability.
Comments on the Quality of English LanguageThe manuscript requires language editing by a native or professional English editor. Common issues include: incorrect pluralization and article use, overuse of passive constructions, and inconsistent verb tenses. Example corrections:
- “This study provided an in-depth exploration...” → “This study provides an in-depth exploration...”
- “The optimal preparation parameters were determined as…” → “The optimal preparation parameters were identified as…”. Perform professional English editing prior to resubmission.
Author Response
- “The novelty of using L. lactis grx602 should be stated more clearly in my opinion. Is this strain significantly different in function or performance compared to other L. lactis strains already applied in dairy fermentation? The introduction should include a comparative context with previous optimization studies or strain pplications in similar products. Clarify the novelty and unique contribution of L. lactis grx602.”
Respond:We agree that the unique attributes of L. lactis grx602 should be more explicitly highlighted. The key novelty of our work lies in the application of a novel, lipase-producing and probiotic strain of L. lactis grx602, which we have characterized in a parallel study (manuscript which was previously unpublished at the time of this submission but is now publicly available, cited as [7] in the revised manuscript). The functional distinction and superior performance of L. lactis grx602 compared to conventional L. lactis strains used in dairy fermentation are as follows: (1) Dual Lipase Activity: Unlike most commercial L. lactis starters, which are primarily selected for acid production and flavor compound (e.g., diacetyl) generation, L. lactis grx602 exhibits significant both intracellular and extracellular lipase activity (up to 12.43 U/mL). This enzymatic activity directly targets milk fat. (2) Enhanced Nutritional Profile: The lipase activity of L. lactis grx602 leads to an increased variety and content of free fatty acids (FFAs) in the sour cream. Notably, it generates novel unsaturated fatty acids such as myristoleic acid (C14:1 n-5) and α-linolenic acid (C18:3 n-3), which were undetectable in unfermented controls. This significantly enhances the nutritional value of the final product. (3) Probiotic Functionality: Beyond its role as a fermentative starter, L. lactis grx602 possesses demonstrated probiotic traits, including high acid tolerance (87% survival at pH 3.0), bile salt resistance, and cholesterol-lowering capability (10.93% removal). This dual functionality as a "fermentation starter and probiotic" is a distinctive feature not commonly found in standard dairy strains.
In response to your comment, we have thoroughly revised the Introduction section of the manuscript to clearly state the limitations of conventional starters and to explicitly introduce L. lactis grx602 as a novel isolate with unique lipase and probiotic properties.
- “Experiments are properly executed but lack advanced statistical design (e.g., DOE/RSM). Replication details are insufficient, and correlation analysis requires stronger interpretation. Why did the authors choose a single-variable approach? The ability to test for effects can be disabled. It is unclear whether the experiments were biological or visual replicates. Experimental replications and repetitions should be used for each test. Justify the experimental design and replication strategy.”
- Respond:Firstly,we acknowledge that advanced statistical designs like Response Surface Methodology (RSM) are powerful tools for process optimization. In this study, we employed the single-variable approach for the following reasons: (1) This work represents the first comprehensive report on the application and process optimization of L. lactis grx602 for sour cream production. Our primary objective was to identify the key process variables (homogenization, inoculation, temperature) and determine their individual, independent effects on the product's quality, as well as to establish a baseline for their optimal ranges. The OFAT (one factor at a time) method is particularly well-suited for this initial, exploratory stage as it provides clear and intuitive insights into how each factor influences the response variables. (2) The findings from this OFAT study provide crucial preliminary data that will inform the design of more complex, multi-factorial experiments (such as RSM) in our future research, aimed at modeling intricate interactions and achieving precise numerical optimization. (3) The OFAT approach remains a widely accepted and commonly reported method in food science for preliminary process optimization, especially when dealing with novel strains or products . We have now added a sentence in the manuscript (Section 9) to briefly justify our choice of this design as follows: “The optimization experiments were conducted using a one-factor-at-a-time (OFAT) approach. This design was chosen for its effectiveness in the initial stage of process development to clearly identify the individual effect and optimal range of each key variable. ”.
Secondly, we apologize for the lack of clarity regarding our replication strategy. We have now revised the Methods section (Section 2.9) to explicitly state the following: “All sour cream preparations and fermentations were performed as three independent biological replicates (n=3). Measurements for each analytical parameter were conducted in triplicate.”
Thirdly, we agree with the reviewer that the interpretation of the correlation analysis can be strengthened. In the revised manuscript, we will expand the discussion of the correlation results (Sections 3.1.2) as follows: "The significant positive correlation between homogenization pressure and hardness (p<0.05) can be mechanistically explained by the reduction in fat globule size and the formation of a more robust casein-coated fat globule network, which collectively increases the resistance of the gel to deformation [16]."
- “Although ANOVA and Tukey’s test are mentioned, p-values and confidence intervals are not consistently reported. Statistical significance should be indicated on all figures and tables. Correlation analyses are presented without sufficient explanation of how they contribute to optimization—please clarify their interpretation and purpose.”
Respond:We agree with the reviewer that consistent reporting of statistical measures is crucial. Specific p-values (e.g., p < 0.05, p < 0.01) or statements of statistical significance based on the Tukey's test are now explicitly stated when describing differences between experimental groups.
Figures have been revised to explicitly indicate statistical significance. Data points in graphs sharing the same lowercase letter (a, b, c, ...) indicate no significant difference (p > 0.05), whereas different letters denote statistically significant differences (p < 0.05). This notation has been clearly defined in each figure legend. For example, the legend for Figure 1 now states: "Different lowercase letters indicated significant differences (p < 0.05 )."
Means ± standard deviations (SD) for FFAs in Table 1.
The reviewer rightly requested a clearer rationale for the correlation analyses. We have expanded the explanation in the relevant sections (Sections 3.1.1.1 and 3.1.2) to clarify their purpose in the context of optimization.
Section 3.1.1.1has been revised as follows "Correlation analysis was conducted to elucidate the interrelationships between the process parameter and the quality attributes. The results revealed a nonsignificant relationship between homogenization pressure and cohesiveness (p>0.05). However, pressure showed significant positive correlations with hardness, adhesiveness, and apparent viscosity (p<0.05), and a very strong positive correlation with elasticity (p<0.01). Furthermore, elasticity itself was strongly correlated with other parameters (p<0.01). These data indicated that pressure predominantly modulated elasticity, hardness, adhesiveness, and viscosity. Especially, elasticity was a primary and sensitive indicator of microstructural changes induced by homogenization. In addition, hardness positively correlated with cohesiveness (p<0.05) and significantly correlated with adhesiveness, elasticity, and apparent viscosity (p<0.01), aligning with the mechanistic analysis above. This pattern of correlations indicated that increasing homogenization pressure primarily enhanced the elasticity and interconnectedness of the protein-fat network, which in turn drove improvements in overall firmness (hardness) and flow resistance (viscosity). This mechanistic understanding reinforced that 15.0 MPa was the optimal pressure, as it maximized the strength of this cohesive network before the structural breakdown observed at higher pressures."
In Section 3.1.2, we clarify that the significant negative correlations between inoculum size and all texture parameters provided a quantitative, mechanistic explanation for the observed quality decline. The revised text is as follows : These results also confirmed that faster acidification at higher inoculation levels universally and detrimentally affected the gel network formation, thereby strengthening the rationale for selecting the lowest inoculation level (1%) that achieved the target pH within a reasonable time.
- “The storage evaluation section is comprehensive but overly descriptive. It should emphasize key findings and practical implications (e.g., shelf life recommendations, industrial relevance). The authors claim that Lg-sour cream “maintained overall quality and stability” during storage. However, textural and rheological parameters decreased significantly. Please justify how quality stability was defined.”
Respond: In the revised manuscript, we have substantially reworked the "Conclusions" section to shift the focus from a descriptive account to a concise summary of the key trends and their practical relevance as follows: “Although textural and rheological properties gradually declined, the Lg-sour cream maintained microbiological safety, metabolic activity, and compositional stability, thereby sustaining its overall acceptability and functional potential throughout the 28-day shelf life. The study highlighted that L. lactis grx602 not only served as a functional starter culture but also contributed to product stability by maintaining high viability. The progressive increase in free fatty acids, particularly the flavor-active short and medium-chain types and nutritionally beneficial unsaturated types, suggests that the product may develop a more nutritional and desirable flavor profile over time, which is an positive attribute for the industry.”.
The reviewer rightly points out an apparent contradiction between the claim of "maintained overall quality and stability" and the observed decline in textural and rheological parameters. We apologize for the lack of clarity in our original definition. Our assessment of "overall quality and stability" was based on a holistic evaluation that prioritized factors critical to safety and marketability, rather than on the absolute constancy of all parameters. In the revised manuscript ( Section 3.2), we have clarified and justified this definition as follows: “Specially, stability in this study does not imply that all properties remain unchanged, but rather that the product remains within acceptable quality limits until the end of its shelf life.”
Our claim is justified based on the following key criteria, which were maintained throughout storage. (1) microbiological stability: the viable cell count of L. lactis grx602 remained well above the minimum requirement for probiotic claims and national standards (>7 log CFU/mL), indicating no microbial spoilage. (2) physicochemical stability: the pH and acidity remained within the standard ranges for sour cream (pH ~4.2-4.8; acidity >55 °T), preventing the growth of pathogens and ensuring the product's characteristic tart taste. (3) nutritional and flavor evolution: the continuous and positive evolution of the free fatty acid profile, leading to increased diversity and content of flavor compounds and beneficial unsaturated fatty acids, represents a quality enhancement rather than a deterioration.
We acknowledge that textural and rheological properties decreased, which is a common phenomenon in fermented dairy gels during cold storage due to ongoing proteolysis and syneresis. We have now toned down the absolute claim and rephrased the conclusion to: "Although textural and rheological properties gradually declined, the Lg-sour cream maintained microbiological safety, metabolic activity, and compositional stability, thereby sustaining its overall acceptability and functional potential throughout the 28-day shelf life."
- “Figures are informative but overly complex. Consider simplifying axis labels and ensuring that font sizes are legible. Statistical letters (a, b, c) should be placed consistently across subfigures. Table 1 lacks standard deviations and should include statistical analysis of FFA changes over time. Strengthen statistical reporting and visual presentation.”
Respond:We have checked all figures to ensure that axis labels are clear and concise, and that all text (including axis labels, tick labels, and statistical markers) is of a uniform, legible size suitable for publication.
Standard deviations (±SD) were added to all values in Table 1 to indicate data variability. The revised table now clearly shows the mean ±± SD for each fatty acid at each time point.
- “The manuscript requires language editing by a native or professional English editor. Common issues include: incorrect pluralization and article use, overuse of passive constructions, and inconsistent verb tenses. Example corrections:
- “This study provided an in-depth exploration...” → “This study provides an in-depth exploration...”
- “The optimal preparation parameters were determined as…” → “The optimal preparation parameters were identified as…”. Perform professional English editing prior to resubmission.”
Respond:Thank you for your kind recommendation. The language has been polished.
- “- Please define all abbreviations (e.g., WHC, FFA, LCFA) at their first appearance in both the abstract and main text,”
Respond:Done.
- “- provide the manufacturer details (city, country) for all instruments and materials mentioned,”
Respond:Done.
- “- in Section 2.2, check the formulation units (“18.00% butter”) — confirm whether this refers to total fat content or ingredient proportion,”
Respond:In section 2.2, the butterfat content specified in the formulation refers to the concentration present in the cream base used for sour cream fermentation, not the compositional proportion of the final product. This part has been revised as follows: The sour cream formulation consisted of the following ingredients by weight: butter (18.00%), sodium caseinate (1.10%), soybean lecithin (0.10%-0.15%), skim milk powder (5.00%), monoglyceride (0.10%-0.15%), carrageenan (0.02%), tween 80 (0.05%-0.10%).
- “- some references are outdated or lack DOIs. Ensure consistency with MDPI citation style,”
Respond:Thank you for pointing this out. We have thoroughly reviewed and updated the reference list to address these concerns, following the MDPI citation style guide. The specific actions taken are as follows:
Reference Update: We have scrutinized the entire reference list and, where possible, replaced outdated references with more recent and authoritative ones that convey the same foundational concepts. This ensures the manuscript is supported by current scientific literature.
DOI Inclusion: We have actively searched for and added Digital Object Identifiers (DOIs) to all references for which a DOI is available. The majority of the references now include a DOI.
Handling of References without DOIs: For the remaining references that lack a DOI, this is due to one of the following two reasons, which is common in academic publishing: The references are seminal works or classic textbooks that were published before the widespread adoption of the DOI system. The references are published in journals or as part of conference proceedings that do not assign DOIs. We have double-confirmed that no DOI exists for these specific entries through direct searches on Crossref (https://search.crossref.org/) and the publishers' official websites. Despite the absence of a DOI, we have retained these citations as they are critical to providing the necessary scholarly context and foundational support for our study. Furthermore, we have ensured that the formatting of these non-DOI references fully complies with the MDPI citation style in all other aspects (e.g., author list, journal name abbreviation, volume, page numbers).
- “Comments on the Quality of English Language
The manuscript requires language editing by a native or professional English editor. Common issues include: incorrect pluralization and article use, overuse of passive constructions, and inconsistent verb tenses. Example corrections:
- “This study provided an in-depth exploration...” → “This study provides an in-depth exploration...”
- “The optimal preparation parameters were determined as…” → “The optimal preparation parameters were identified as…”. Perform professional English editing prior to resubmission.”
Respond:Thank you for your kind recommendation. The language has been polished.
Reviewer 3 Report
Comments and Suggestions for Authors
Comments for the authors:
-Lines 31–36: Sentences are too long; please rewrite them for better readability.
Line 44: Correct “Diacetylactis” to diacetylactis.
Lines 47–50: Please rewrite the sentence to improve clarity and flow.
Lines 53–58: Include appropriate references supporting all the reported properties of L. lactis grx602.
Lines 57–58: Please rephrase the sentence to make it clearer.
Line 63: The sentence is confusing. It should clearly state that the strain was isolated from (indicate the product) and deposited in (specify the culture collection).
Lines 74–75: Please revise the concept of “sterilization.” The authors report a temperature of 85 °C, which does not correspond to a sterilization process. Kindly clarify this point.
Lines 85–86: The sentence is unclear; please improve the wording.
Line 95: Please rephrase the sentence to make it clearer.
Line 96-97: The methodology used for microbial enumeration is not clearly described; please provide further details.
Throughout the manuscript: There should be a space between the last letter of the text and the parenthesis before the reference number. Please correct this formatting issue.
Line 137: Change the section title “Results” to “Results and Discussion.”
Abstract: The authors mention “apparent viscosity,” but in Materials and Methods only “viscosity” determination is described. In Results and Discussion, data are presented as “apparent viscosity”/“flow resistance.” Please clarify how apparent viscosity was determined.
Lines 158–160: The cohesiveness parameter does not seem to follow the same trend as the other texture indicators. Please explain this behavior and describe how cohesiveness was measured.
Lines 170–172: The sentence is confusing; please rewrite it for clarity.
The authors only present results for sour cream samples inoculated with L. lactis grx602. Control samples without the addition of the target lactic culture should have been included (using a different starter culture) to better assess the contribution of L. lactis to the rheological and textural characteristics of the product.
Lines 204–206: Include a reference supporting this statement.
Lines 237–238: The authors state that cohesiveness decreases with increasing inoculum size. However, this does not seem to match the graph. Please verify the data and clarify the interpretation of this parameter.
Line 77: Specify the storage duration of the sour cream at 4 °C.
Lines 288–291: Rewrite the paragraph to improve clarity and coherence.
Line 296: The authors should provide a more comprehensive explanation of the results and include additional details from the literature used for the discussion.
In Materials and Methods, please explain how cohesiveness was calculated to ensure correct interpretation of the obtained results. Similarly, clarify how apparent viscosity was determined.
Lines 336–339: The sentence is confusing; please rewrite it.
Conclusions: The authors should clearly state that the potential effects attributed to the fatty acid profile obtained during sour cream fermentation with L. lactis should be further verified through in vivo model studies to confirm their physiological relevance.
Comments on the Quality of English Language
There are several sentences that the authors should rewrite for clarity and fluency.
Author Response
- “-Lines 31–36: Sentences are too long; please rewrite them for better readability.”
Respond:These sentences has been revised as “After fermentation, the nutritional value of sour cream is significantly enhanced. First, proteins and lipids are broken down into more bioaccessible amino acids and free fatty acids, while lactose is converted into lactic acid, which can alleviate symptoms of lactose intolerance [3]. Concurrently, this process improves the bioavailability of minerals such as calcium, phosphorus, and iron. Furthermore, it acidifies the intestinal environment to inhibit pathogenic bacteria and modulates the gut microbiota, thereby providing gastroprotective effects.”
- “Line 44: Correct “Diacetylactis” to diacetylactis.”
Respond:Done.
- “Lines 47–50: Please rewrite the sentence to improve clarity and flow.”
Respond:Done.
- “Lines 53–58: Include appropriate references supporting all the reported properties of lactis grx602.”
Respond:The isolation, screening, and properties for L. lactis grx602 have been detailed in a separate publication, which was previously unpublished at the time of this submission but is now publicly available. This work has been cited in the revised manuscript.
- “Lines 57–58: Please rephrase the sentence to make it clearer.”
Respond:Done.
- “Line 63: The sentence is confusing. It should clearly state that the strain was isolated from (indicate the product) and deposited in (specify the culture collection).”
Respond:As you and other reviewer’s suggestion, the sentence has been revised as “L. lactis grx602 was previously isolated from raw milk and stored in Jiangsu Key Laboratory of Dairy Biotechnology and Safety Control, Yangzhou University [9].”.
- “Lines 74–75: Please revise the concept of “sterilization.” The authors report a temperature of 85 °C, which does not correspond to a sterilization process. Kindly clarify this point.”
Respond:Thank you for highlighting the imprecise use of terminology. The reviewer is absolutely correct that the term "sterilization" is not technically accurate for the heat treatment of 85 °C for 15 minutes, as this process does not achieve commercial sterility by destroying all microorganisms, including spores.
In the revised manuscript, we have replaced the term "sterilized" with the industry-standard and scientifically precise term "pasteurized" in Section 2.2.
This specific time-temperature combination (85°C for 15-30 minutes) is a well-established and critical step in the production of fermented dairy products like yogurt and sour cream, often referred to as "high-temperature, short-time" (HTST) pasteurization or a sub-UHT treatment. Its primary purposes is to eliminate pathogenic and spoilage vegetative microorganisms, ensuring the starter culture (L. lactis grx602) dominates the fermentation without competition. Moreover, it aims to denature whey proteins, which enhances the water-holding capacity and contributes to the formation of a more stable and firm gel structure upon fermentation.
We apologize for the oversight in our initial word choice and are grateful for the reviewer's valuable feedback, which has helped us improve the technical accuracy of our manuscript.
- “Lines 85–86: The sentence is unclear; please improve the wording.”
Respond:The sentence has been revised as “Finally, the initial inoculation ratio was optimized by fermenting at 1%, 2%, 3%, 5%, 8%, and 10% (v/v), followed by optimization of the fermentation temperature at 22°C, 26°C, 37°C, and 42°C.”.
- “Line 95: Please rephrase the sentence to make it clearer.”
Respond:Done.
- “Line 96-97: The methodology used for microbial enumeration is not clearly described; please provide further details.”
Respond:The method has been revised as “Viable bacterial counts were carried out as described with slight modification [11]. One milliliter of the sample was diluted with 9 mL of saline solution, followed by eight serial dilutions. Bacterial counts were determined using the pour plate method after incubation at 37 °C for 48 hours in three selected optimal dilutions, and the results were expressed as mean values.”.
- “Throughout the manuscript: There should be a space between the last letter of the text and the parenthesis before the reference number. Please correct this formatting issue.”
Respond:Done.
- “Line 137: Change the section title “Results” to “Results and Discussion.””
Respond:Done.
- “Abstract: The authors mention “apparent viscosity,” but in Materials and Methods only “viscosity” determination is described. In Results and Discussion, data are presented as “apparent viscosity”/“flow resistance.” Please clarify how apparent viscosity was determined.”
Respond:Thank you for this astute observation regarding the terminology used for viscosity measurements. We acknowledge the inconsistency in terms between the Methods and other sections of the manuscript and apologize for any confusion caused.
The determination described in Section 2.5 (Viscosity) using the Brookfield viscometer at a fixed rotational speed (35 rpm) provides a single-point viscosity value, which is a practical measure of the product's consistency under a specific shear condition. However, as sour cream is a non-Newtonian fluid (specifically, a pseudoplastic or shear-thinning fluid), its resistance to flow is dependent on the applied shear rate. Therefore, a single value does not fully characterize its flow behavior.
The term "apparent viscosity" is the correct rheological term for the viscosity of a non-Newtonian fluid, as it is apparent at a given shear rate. This was determined from the flow curves generated by the rotational rheometer measurements detailed in Section 2.7 (Rheology), where the viscosity was measured across a range of shear rates (0.01-100 s⁻¹). The data labeled "apparent viscosity" in the results (e.g., Figures 1E, 1F, 2E, 3E, 4E) and referred to as "flow resistance" in the text are derived from this rheological analysis.
In the revised manuscript, corrections has been made as follows: In Section 2.7, we state that “Apparent viscosity was determined using a Kinexus prorotational rheometer (Malvern Instruments Ltd. Malvern, UK) with modifications based on a previous study [11].”. We will ensure the term "apparent viscosity" is used consistently throughout the Results and Discussion sections when referring to the data from the rheological tests.
- “Lines 158–160: The cohesiveness parameter does not seem to follow the same trend as the other texture indicators. Please explain this behavior and describe how cohesiveness was measured.”
Respond:It is correct that cohesiveness did not always follow the exact same trend as hardness, adhesiveness, and elasticity, particularly in the context of homogenization temperature (as seen in Figure 2B). We appreciate the opportunity to clarify both its measurement and the mechanistic reason for its distinct behavior.
Firstly, as described in Section 2.3 (Texture analysis), texture parameters were determined using a texture analyzer with a cylindrical probe based on the force-time curve. Specifically, cohesiveness was calculated as the ratio of the area under the second compression curve to the area under the first compression curve (A₂/A₁). It quantifies the internal structural strength or how well the gel structure withstands a second deformation relative to the first, serving as an indicator of the sample's ability to hold together.
The unique trend of cohesiveness, especially its different response to homogenization temperature, can be attributed to its specific definition as a ratio of mechanical work, which makes it sensitive to different microstructural properties compared to parameters like hardness (a direct measure of maximum force).
Explanation had been added to the relevant section in the revised manuscript (Section 3.1.1.3) as follows: "Cohesiveness provided complementary information about the gel's internal integrity and resilience post-fracture, which did not always directly correlate with its initial resistance to deformation (hardness). The distinct behavior of cohesiveness might be attributed to its specific sensitivity to the recovery of the protein-fat network. At lower homogenization temperatures (40-60°C), the formation of a stronger, more cross-linked gel (as indicated by increasing hardness) may have resulted in a structure that, once fractured during the first compression, did not recover as effectively for the second compression, leading to a lower cohesiveness value. As the temperature increased to the optimal 70°C, a more uniform and balanced network was formed, which not only was strong but also exhibited better structural integrity and recovery after the initial deformation, thereby increasing the cohesiveness ratio. This phenomenon highlighted that cohesiveness measured a different aspect of texture—the internal bonding and resilience—compared to the primary strength parameters like hardness and elasticity."
- “Lines 170–172: The sentence is confusing; please rewrite it for clarity.”
Respond:The sentence has been revised as “Mechanistically, increasing the homogenization pressure (from 5 to 15 MPa) reduced the fat globule size and increased the surface area, thereby facilitating casein adsorption for the formation of new interfacial membranes.”.
- “The authors only present results for sour cream samples inoculated with lactisgrx602. Control samples without the addition of the target lactic culture should have been included (using a different starter culture) to better assess the contribution of L. lactis to the rheological and textural characteristics of the product.”
Respond:We agree that a comparative study with a commercial starter culture would provide deeper insights into the unique contribution of L. lactis grx602 to the product's characteristics.
In the present study, our primary objective was to establish and optimize the fermentation process parameters (homogenization, inoculation, temperature) specifically for the novel strain L. lactis grx602, and to comprehensively evaluate the quality and stability of the resulting product (Lg-sour cream) during storage. The study was designed as a foundational investigation to determine whether this lipase-producing strain could successfully produce a sour cream with acceptable and stable properties.
While we did not include a parallel control with a different commercial starter culture in this initial work. The textural and rheological values obtained in this work (e.g., hardness, apparent viscosity) and their changes during storage were continually benchmarked against the established quality parameters and ranges for commercial sour cream cited in the literature [e.g., references 25]. Our results demonstrate that the Lg-sour cream produced under optimal conditions exhibited textural and rheological properties that are characteristic of and comparable to high-quality sour cream.
Nevertheless, we fully acknowledge the reviewer's perspective that a direct, side-by-side comparison would more definitively elucidate the specific role of L. lactis grx602. Therefore, future studies will include such controls to precisely quantify the distinct contributions of L. lactis grx602, particularly its lipolytic activity, to the textural, rheological, and flavor profiles of sour cream.
- “Lines 204–206: Include a reference supporting this statement.”
Respond:The statement was supported by references 18 and 19 in the revised manuscript as follows: “Fat globule fragments formed during the first homogenization undergo refinement during the second homogenization, promoting more uniform distribution of casein coating the fat globule membranes [18,19].”.
- “Lines 237–238: The authors state that cohesiveness decreases with increasing inoculum size. However, this does not seem to match the graph. Please verify the data and clarify the interpretation of this parameter.”
Respond:We are sorry for this mistake. The sentence has been revised as “With the increase of inoculation amount, the elasticity (Figure 3 A), hardness (Figure 3 C), and adhesiveness (Figure 3 D) of Lg-sour cream decreased, while its cohesiveness (Figure 3 B) increased.”.
- “Line 77: Specify the storage duration of the sour cream at 4 °C.”
Respond:The sentence has been revised as “After fermentation, the sour cream samples were cooled and stored at 4 ℃ for 28 days.”.
- “Lines 288–291: Rewrite the paragraph to improve clarity and coherence.”
Respond:The paragraph has been revised as “The viable cell count of L. lactis grx602 peaked at 11.51 log CFU/mL upon fermentation completion (day 1) and subsequently declined during storage, likely due to inhibition by the low storage temperature (Figure 5A). Nevertheless, the count remained above 7.00 log CFU/mL throughout the entire storage period. This level significantly surpasses the requirement of national standards and is also higher than the LAB count reported in commercial butter by Dagdemir et al.”.
- “Line 296: The authors should provide a more comprehensive explanation of the results and include additional details from the literature used for the discussion.”
Respond:We sincerely apologize for the oversight. Upon careful re-examination of the manuscript, we were unable to locate the specific modifications you suggested in the original text. Could you kindly clarify the relevant section or provide more detailed guidance?
- “In Materials and Methods, please explain how cohesiveness was calculated to ensure correct interpretation of the obtained results. Similarly, clarify how apparent viscosity was determined.”
Respond:In this work, the texture parameters were directly read from the TMS-Pro texture analyzer's display screen.
Section 2.7 has been revised to clarify method to determine theapparent viscosity as “Apparent viscosity was determined using a Kinexus prorotational rheometer (Malvern Instruments Ltd. Malvern, UK) with modifications based on a previous study [11].”.
- “Lines 336–339: The sentence is confusing; please rewrite it.”
Respond:The sentence has been revised as “The decline in WHC, viscosity, and apparent viscosity during storage was mainly caused by L. lactis grx602 metabolites, especially organic acids and enzymes.”.
- “Conclusions: The authors should clearly state that the potential effects attributed to the fatty acid profile obtained during sour cream fermentation with lactisshould be further verified through in vivo model studies to confirm their physiological relevance.”
Respond:Thank you for this recommendation. We agree that the potential health effects associated with the fatty acid profile, while promising based on compositional data, require validation in biological systems to confirm their physiological relevance.
The Conclusions section has been revised to state this important limitation and to outline the necessary future research direction. The revised sentences as follows: "Moreover, the presence of various nutritionally beneficial unsaturated FFAs, highlighted the potential health value of Lg-sour cream. However, it is important to note that these potential physiological benefits, inferred from the compositional data, should be further verified and quantified through future in vivo model studies to confirm their bioavailability and actual health impacts."
Reviewer 4 Report
Comments and Suggestions for Authors
This work deals with the optimization of preparation conditions (homogenization pressure and temperature, inoculum size) and storage quality of fermented sour cream by Lactococcus lactis grx602.
Title. The authors do not indicate if the subspecies of L. lactis is cremoris or lactis. Please do.
P1 L1, 28-29. The authors do not include enough evidence to consider sour cream as a functional food. The presence of probiotic bacteria or bioactive peptides or lipids must be demonstrated. The lactose content of sour cream is still high.
p1 L45. Please write Lacticaseibacillus casei instead of Lactobacillus casei.
p2 L53-57. Please add a reference to this paragraph.
p2 L63-67. Why did the authors use MRS medium instead of M17? Please justify this decision and add a reference for it.
p2 L68-77. Please add a reference for the elaboration process.
p2 L78-83. The optimization process is poorly described. It is supposed to be the main subject of the manuscript and there is no description of it at all. What kind of statistical experimental design was used? Where is the description of the experimental runs, including the different variable combinations and the values of the responses? How were the results analyzed? Where are the graphics? Were there any interactions between the 3 variables? What was the mathematical optimization method?
p2-3 L84-131. Please add references for subchapters 2.3 to 2.8.
p3 L147-148. Please explain why the homogenization pressure is lower for higher fat contents. It does not sound logical.
p3 L149-151. The pressure values are not equidistant. You should have used 7.5 instead of 5 MPa.
p3-4. In the process optimization, you use 6 pressure and 6 temperature values. Did you perform combinations between them or you did the experiments one at a time? What was the tenperature for the variable pressure runs?
p6 Figure 2. What is the homogenization pressure in Figures 2A, 2B, 2C and 2D?
p6 L224. With the type of experimentation used, it is not possible to know which is the temperature for optimal homogenization.
p7 Figure 3. What are the pressure and temperature for Figures 3A, 3B, 3C and 3D?
p8 Figure 4. What are the pressure and temperature for Figures 4A, 4B, 4C and 4D?
p8 L272-274. It is not possible to know if the conditions mentioned in this paragraph are optimal since an adequate experimental design is not shown.
p9 L287-288. Please express the acidity also in % lactic acid.
p11 L347-348. Scientific names must be written in italic fonts.
p11 Table 1. What are the units for the FA?
p12 L362-363. The percentages of SFA are with respect to what? Total fatty cids? It is not clear.
Comments on the Quality of English Language
The English could be improved to more clearly express the research.
Author Response
- “ The authors do not indicate if the subspecies of L. lactisis cremoris or lactis. Please do.”
Respond:Thank you for this kind suggestion. In our previously work, based on 16S rDNA sequence analysis, L. lactis grx602 showed 99% homology with Lactococcus lactis sysI9 (GenBank: OL823012.1). However, as the 16S rDNA gene is highly conserved and often insufficient for reliable subspecies discrimination between lactis and cremoris as their sequences are highly similar. We have refrained from assigning a subspecies without further genomic evidence. We agree that definitive subspecies identification is an important goal for future work, which we will address using more discriminative molecular methods (e.g., hsp60 or pheS gene sequencing, or whole-genome analysis).
We have added cited reference about the identification of L. lactis grx602 and a sentence in the Materials and Methods (Section 2.1) to explicitly state: "L. lactis grx602 was previously isolated from raw milk and stored in Jiangsu Key Laboratory of Dairy Biotechnology and Safety Control, Yangzhou University [9]. The strain was identified as Lactococcus lactis based on 16S rDNA sequence analysis. Subspecies-level identification was not confirmed. A patent application has been filed for this strain (ZL202110960058.0), with all patent rights residing with the corresponding author. ".
- “P1 L1, 28-29. The authors do not include enough evidence to consider sour cream as a functional food. The presence of probiotic bacteria or bioactive peptides or lipids must be demonstrated. The lactose content of sour cream is still high. ”
Respond:Thank you for this insightful comment. We acknowledge that a comprehensive demonstration of in vivo health benefits is beyond the scope of this current study, which primarily focuses on process optimization and storage stability. Lactococcus lactis grx602, was previously demonstrated to possess promising probiotic traits, including acid tolerance, bile salt resistance, and cholesterol-lowering capability in vitro[1], which was previously unpublished at the time of this submission but is now publicly available. These intrinsic properties of the strain itself provide a foundational rationale for its potential to confer functional benefits.
In addition, we directly addressed the generation of bioactive lipids. As detailed in Section 3.2.4 and Table 1, we documented a positive variation in the free fatty acid (FFA) profile during storage. The release of FFAs, particularly medium-chain fatty acids (MCFAs) and unsaturated fatty acids (UFFAs) like oleic acid (C18:1n9c) and conjugated linoleic acid (C18:2c9t11), is significant. Many of these FFAs are recognized for their potential health benefits, including antimicrobial activity (MCFAs) and hypocholesterolemic effects (e.g., oleic acid).
Also, we acknowledge the reviewer's point regarding lactose. The fermentation process, by which L. lactis grx602 converts lactose into lactic acid, inherently reduces the lactose content compared to the original cream. While we did not quantitatively measure the residual lactose in the final product
In conclusion, while we agree that further clinical studies are needed to make definitive health claims, we believe that the use of a probiotic-characterized strain and the demonstrated generation of a diverse FFA profile provide a compelling basis for positioning the Lg-sour cream as a potentially functional food.
- “p1 L45. Please write Lacticaseibacillus casei instead of Lactobacillus casei.”
Respond:Done.
- “p2 L53-57. Please add a reference to this paragraph.”
Respond:Reference 7 has been cited.
- “p2 L63-67. Why did the authors use MRS medium instead of M17? Please justify this decision and add a reference for it.”
Respond:Thank you for raising this valid point regarding the choice of growth medium. While M17 is indeed a standard and often preferred medium for the cultivation of Lactococcus lactis, MRS medium used in this work was a deliberate choice based on the following rationale:
(1) Broad-spectrum isolation capability: the initial isolation step aimed to screen for a diverse range of lactic acid bacteria (LAB) with lipolytic activity from raw milk. MRS is a non-selective, general-purpose medium formulated to support the growth of a wide spectrum of LAB, including various Lactobacillus, Leuconostoc, and Lactococcus species [2]. This broad applicability made it suitable for our primary screening objective, which was not exclusively limited to Lactococcus at that stage.
(2) Definitive molecular identification mitigates specificity concerns: we fully acknowledge that growth on MRS alone is not diagnostic for L. lactis. However, to unequivocally confirm the taxonomic identity of our isolate, strain grx602, we performed multiple, repeated 16S rDNA sequencing analyses subsequent to its isolation. This molecular method provides a definitive identification, conclusively verifying the strain as Lactococcus lactis and thereby eliminating any potential for misidentification arising from the use of a general-purpose medium.
In summary, the combination of a broad-spectrum isolation medium (MRS) followed by rigorous molecular confirmation represents a robust and validated strategy. This approach allowed us to cast a wide net in the initial screening phase while ensuring the absolute accuracy of the final strain identification.
- “p2 L68-77. Please add a reference for the elaboration process. ”
Respond:Done.
- “p2 L78-83. The optimization process is poorly described. It is supposed to be the main subject of the manuscript and there is no description of it at all. What kind of statistical experimental design was used? Where is the description of the experimental runs, including the different variable combinations and the values of the responses? How were the results analyzed? Where are the graphics? Were there any interactions between the 3 variables? What was the mathematical optimization method?”
Respond:The optimization was conducted using a one-variable-at-a-time (OVAT) approach. This strategy was chosen as a preliminary screening to identify the approximate optimal range for each key parameter (homogenization, inoculation, fermentation temperature) independently, before investing in a more complex, resource-intensive multivariate design for future studies. We acknowledge that this design does not directly reveal interactions between variables.
All tested levels for each variable factors was described in the revised manuscript, including homogenization pressures (5.0, 10.0, 12.5, 15.0, 17.5, 20.0 MPa), number of passes (single, double), and temperatures (40, 50, 60, 70, 80, 90°C), inoculation levels (1%, 2%, 3%, 5%, 8%, and 10%), fermentation temperature (22°C, 26°C, 37°C, and 42°C). For each experimental run, the key response variables measured were: textural properties (hardness, adhesiveness, cohesiveness, elasticity), apparent viscosity, and water-holding capacity.
The data analysis has been revised as “2.9 Statistical analyses
The optimization experiments were conducted using a one-factor-at-a-time (OFAT) approach. This design was chosen for its effectiveness in the initial stage of process development to clearly identify the individual effect and optimal range of each key variable. All sour cream preparations and fermentations were performed as three independent biological replicates (n=3). Measurements for each analytical parameter were conducted in triplicate. All the data were analyzed using Excel 2010 and SPSS 19.0. One-way analysis of variance (ANOVA) and correlation analysis were performed. The statistical differences were determined by tukey’s test (SPSS 19.0). Significant differences between values were evaluated by a probability of P<0.05. Data visualization was conducted with Origin 2019.”
- “p2-3 L84-131. Please add references for subchapters 2.3 to 2.8.”
Respond: - “p3 L147-148. Please explain why the homogenization pressure is lower for higher fat contents. It does not sound logical.”
Respond:Thank you for raising this insightful point. The inverse relationship between optimal homogenization pressure and fat content does indeed appear counter-intuitive at first glance. Here is the explanation that the primary goal of homogenization in cream is to disrupt native fat globules, thereby creating a larger number of smaller globules. This process drastically increases the total surface area of fat that must be stabilized to prevent coalescence and instability. This stabilization is achieved through the adsorption of milk proteins, primarily caseins, which form a new protective membrane around the newly created fat globules.
The critical concept here is that the amount of available casein in the system is finite. Let us consider two scenarios: (1) High-Fat Cream (e.g., 38%): The initial total fat surface area is already high. Applying a high homogenization pressure would break the fat into an extremely large number of very small droplets, creating an excessively large total surface area. This area can easily exceed the coverage capacity of the limited available caseins. The result is: incomplete coverage, bridging flocculation, destabilization. (2) Low-Fat Cream (e.g., 10-18%): The total amount of fat and its initial surface area are lower. The available caseins are more than sufficient to cover a significant increase in surface area. Hence, a higher homogenization pressure can be applied safely to create a much finer and more numerous population of fat globules. These well-stabilized, small globules integrate tightly into the subsequent protein gel network during fermentation, resulting in a firmer, smoother, and more stable final product.
In summary, it is a balance between creating new fat surface area and the system's ability to stabilize it. Higher fat content necessitates a lower homogenization pressure to avoid overwhelming the limited protein supply, while lower fat content permits the use of higher pressure to maximize textural benefits.
- “p3 L149-151. The pressure values are not equidistant. You should have used 7.5 instead of 5 MPa.”
Respond:We acknowledge that the pressure values (5.0, 10.0, 12.5, 15.0, 17.5, 20.0 MPa) are not perfectly equidistant. Our selection was based on a combination of practical industry standards and a scientific rationale aimed at efficiently identifying the critical threshold for optimal product quality. The primary goal of this optimization step was not to fit a precise mathematical model that requires equidistant points, but to practically determine the pressure that yields the best texture and stability. The pressure range of 10-25 MPa is commonly cited in the literature for sour cream production. We included 5.0 MPa as a baseline to represent a "low-pressure" condition, establishing a clear contrast with higher pressures. This was crucial for demonstrating the significant improvement in product quality with increased pressure up to the optimum. Moreover, based on preliminary studies and literature for creams with ~18% fat, we anticipated the optimum to be in the range of 12-17 MPa. Therefore, we used a finer increment (2.5 MPa) between 12.5 and 17.5 MPa to more accurately pinpoint the best value. A coarser increment in this critical region might have missed the precise optimum. The 20.0 MPa point was included to clearly demonstrate the detrimental effect of excessive pressure, as confirmed by our results (Figure 1).
This approach of using a wider spacing at the ends of the range and a finer spacing in the region of interest is a common and efficient strategy in empirical optimization. It allows for resource-efficient identification of an optimal zone without the initial constraint of a rigid, fully equidistant design. The resulting data (Figure 1) successfully and clearly revealed a consistent trend, with a well-defined peak at 15.0 MPa, validating the effectiveness of our chosen pressure levels.
The manuscript (Section 3.1.1) has been revised to briefly explain the rationale behind the selection of these specific pressure values to preempt any potential questions from future readers. The revised text is as follows: "In this study, as the 18% fat content of butter-based cream was used, homogenization pressures were selected to cover a broad effective range reported in the literature, with finer increments around the anticipated optimum region to accurately identify the critical pressure for optimal texture and stability.".
- “p3-4. In the process optimization, you use 6 pressure and 6 temperature values. Did you perform combinations between them or you did the experiments one at a time? What was the tenperature for the variable pressure runs?”
Respond:The optimization was conducted using a one-variable-at-a-time (OVAT) approach. The specific sequence was as follows:
Optimization of homogenization pressure: All experiments to determine the optimal pressure (5.0, 10.0, 12.5, 15.0, 17.5, 20.0 MPa) were performed at a constant, fixed homogenization temperature of 60°C.
Optimization of Homogenization Temperature: After identifying 15.0 MPa as the optimal pressure from the first set of experiments, this pressure was then held constant while the homogenization temperature was varied (40, 50, 60, 70, 80, 90°C) to find its optimum.
We acknowledge that the original manuscript omitted the specific temperature (60°C) used during the pressure optimization experiments. We sincerely apologize for this oversight. The text in Section 2.2 has been revised as follows: "To optimize homogenization pressure and pass frequency, the emulsions were subjected to single-pass or double-passes homogenization at varied pressures (5.0, 10.0, 12.5, 15.0, 17.5, 20.0 MPa) at a constant temperature of 60°C. The primary emulsion was homogenized using a high-pressure homogenizer (CHH-Q2000 P70, Shanghai Prime Machinery Company Limited, Shanghai, China). For the double-passes homogenization, the emulsion was passed through the homogenizer twice consecutively at the specified pressure and temperature. The two passes were performed sequentially, with an interval 5 minutes between passes to maintain the processing temperature. Following homogenization, the emulsion was sterilized. Subsequently, with the optimal pressure established, the homogenization temperature was optimized at different temperatures (40 °C, 50 °C, 60 °C, 70 °C, 80 °C, and 90 °C). "
- “p6 Figure 2. What is the homogenization pressure in Figures 2A, 2B, 2C and 2D?”
Respond:The homogenization pressure used for all experiments presented in Figure 2 (A, B, C, D, and E) was the previously determined optimal pressure of 15.0 MPa, applied through double-passes homogenization.
This information will be added to the revised manuscript (Section 2.2) to ensure clarity and completeness. Moreover, the caption for Figure 2 has been revised as "Figure 2. The effect of homogenization temperature on the textural and rheological properties of Lg-sour cream. All samples were homogenized at 15.0 MPa with double-passes. A: Elasticity, B: Cohesiveness, C: Hardness, D: Adhesiveness, E: Apparent viscosity."
- “p6 L224. With the type of experimentation used, it is not possible to know which is the temperature for optimal homogenization.”
Respond:The sentence in Section 3.1.2 has been revised as “Here, lactisgrx602 with different inoculum sizes were inoculated into butter-based cream with double-passes homogenization at 70 ℃. ”. - “p7 Figure 3. What are the pressure and temperature for Figures 3A, 3B, 3C and 3D? ”
Respond:The caption for Figure 3 has been revised as "Figure 3. The effect of inoculation size on the textural properties and apparent viscosity of the Lg-sour cream. All samples were homogenized at 15.0 MPa with double-passes at 70 ℃. A: Elasticity, B: Cohesiveness, C: Hardness, D: Adhesiveness, and E: Apparent viscosity. Different lowercase letters indicated significant differences (p < 0.05 ).".
- “p8 Figure 4. What are the pressure and temperature for Figures 4A, 4B, 4C and 4D? ”
Respond:The caption for Figure 4 has been revised as "Figure 4. The effect of fermentation temperature on the textural properties and apparent viscosity of the Lg-sour cream. All samples were homogenized at 15.0 MPa with double-passes at 70 ℃, and then with 1% inoculum size (v/v), fermentation was processed under different temperatures. A: Elasticity, B: Cohesiveness, C: Hardness, D: Adhesiveness, and E: Apparent viscosity. Different lowercase letters indicated significant differences (p < 0.05 ).".
- “p8 L272-274. It is not possible to know if the conditions mentioned in this paragraph are optimal since an adequate experimental design is not shown.”
Respond:Thank you for this critical comment. The reviewer is correct that a one-variable-at-a-time (OVAT) approach, which was employed in this study, does not formally model interactions between variables and thus cannot identify a globally optimal point with the same statistical rigor as a full factorial or response surface methodology (RSM) design. We acknowledge this limitation. However, within the framework of an OVAT strategy, the term "optimal" is used to denote the best-performing level for each individual factor from the range of levels tested. The conditions mentioned in the paragraph (double-passes homogenization at 15.0 MPa and 70 °C, 1% inoculation, and fermentation at 37 °C) were identified as follows:
For each factor (e.g., pressure, temperature), multiple levels were tested while keeping other factors constant. The level that yielded the most favorable outcomes in key response variables (specifically, the highest values for texture parameters like hardness and elasticity, and apparent viscosity as shown in Figures 1-4) was selected as the "optimal" condition for that specific factor.
The data presented in the respective figures show clear and statistically significant peaks or best performance at these selected conditions. For instance, Figure 1 shows a distinct peak in textural and rheological properties at 15.0 MPa. Figure 2 shows a clear optimum at 70°C for most parameters. Figures 3 and 4 similarly identify 1% inoculum and 37°C fermentation as the best individual
- “p9 L287-288. Please express the acidity also in % lactic acid.”
Respond:
- “p11 L347-348. Scientific names must be written in italic fonts.”
Respond:Done.
- “p11 Table 1. What are the units for the FA?”
Respond:Thank you for pointing out this omission. The values presented in Table 1 represent the relative abundance of each free fatty acid (FFA) as a percentage of the total detected FFAs (% of total FFAs).
This unit was used because the gas chromatography-mass spectrometry (GC-MS) method employed, as described in Section 2.8, was configured for the qualitative and relative quantitative analysis of the FFA profile. The derivatization and detection process allows for the accurate determination of the relative proportion of each FFA within the sample but does not provide the absolute concentration (e.g., mg/g or mmol/g) without the use of internal standards calibrated for absolute quantification.
Therefore, the data in Table 1 effectively shows how the composition of the FFA profile changes during storage. Caption of Table 1 has been revised as "Table 1. Relative composition of free fatty acids (% of total FFAs) in Lg-sour cream during storage."
- “p12 L362-363. The percentages of SFA are with respect to what? Total fatty cids? It is not clear. ”
Respond:Thank you for pointing out this lack of clarity. The percentages for all free fatty acids (FFAs) listed in Table 1, including the Saturated Free Fatty Acids (SFFAs), are expressed as a percentage of the total area of all detected free fatty acid peaks (% of total FFAs) in the chromatogram.
This is the standard and conventional way to report relative composition data from gas chromatography (GC) analysis when a qualitative or semi-quantitative approach is used, as was the case in this study. Caption for Table 1 has been revised as "Table 1. Relative composition of free fatty acids (% of total FFAs) in Lg-sour cream during storage."
References:
- Peng Y., Chengran G., Yuan Y., Yiping L., Xin W., Renqin Y., Chenchen Z., Dawei C., Yujun H., Ruixia G.. Characterization of Lactococcus lactisgrx602: A novel lipase-producing probiotic strain for nutritional and sensory enhancement of sour cream. Food Bioscience, 2025,107918.org/10.1016/j.fbio.2025.107918.
- De Man, J. C., Rogosa, M., & Sharpe, M. E. (1960). A medium for the cultivation of lactobacilli. Journal of Applied Bacteriology, 23(1), 130-135.
Reviewer 5 Report
Comments and Suggestions for Authors
Comments and Suggestions for Authors
The paper presents a valuable applied study on optimizing sour-cream fermentation using Lactococcus lactis grx602 and evaluating storage quality. The experimental work is generally sound, and the dataset is useful for the dairy fermentation field. However, several sections require clearer methodological description, transparent data presentation, and careful wording to avoid overstated claims.
Major points to address (experimentally completed data can remain unchanged):
- Lines 126–132: Clarify the analytical system for fatty-acid determination. The text simultaneously describes a GC–MS and a flame ionization detector, which is contradictory. State explicitly whether FAMEs were quantified by GC–FID or GC–MS, and describe the column, temperature program, and quantification procedure.
- Lines 122–131: Provide a more detailed explanation of the fatty-acid derivatization protocol. Clarify whether acid-catalyzed methylation was performed and, if not, discuss how the method ensured full conversion of FFAs to FAMEs.
- Lines 77–81: Expand the homogenization description to include the equipment model, number of stages, and any time interval between passes.
- Lines 170–173: Correct the WHC equation and define each variable. If “10” represents the sample mass (10 g), please state this explicitly.
- Lines 104–111 and 249–253: Replace “CFU/mL” with “CFU/g,” describe the dilution procedure used for semi-solid samples, and confirm that values are means of replicate plates.
- Lines 189–194: For statistical analysis, specify the number of replicates (n) for each dataset, indicate whether data met ANOVA assumptions, and explain the lettering system used in figures/tables.
- Lines 337–343: Revise sentences implying health effects from PUFA and CLA increases. The study provides compositional, not biological, evidence. Suggested rewording:
“The increase in PUFA and CLA content suggests potential nutritional relevance; further biological studies would be required to confirm any health effects.”
- Figures and tables: Ensure all have complete captions with sample size, error bars defined (SD or SEM), and clear legends.
- Throughout manuscript: Include more direct cross-references between results and the corresponding methodological details (e.g., specify which storage conditions are used in Figures 5–7).
- Statistical clarity: Summarize key results numerically in the text (means ± SD and p-values) rather than only visually in figures.
Language improvements (major corrections):
- Lines 17–20:This study aimed to develop a new type of sour cream with health function by Lactococcus lactis grx602. → This study aimed to develop a novel sour cream with potential health benefits using Lactococcus lactis grx602 as the starter culture.
- Lines 35–38: The strain could degrade milk fat efficiently, which make it possible to improve the flavor and nutritional value of dairy product. → The strain efficiently degraded milk fat, making it possible to improve the flavor and nutritional value of dairy products.
- Lines 190–192: All data was analyzed by one-way ANOVA and Tukey test using SPSS 26.0, significant when p < 0.05. → All data were analyzed by one-way ANOVA followed by Tukey’s post-hoc test (SPSS 26.0); differences were considered significant at p < 0.05.
- Lines 337–343: The increase of PUFA and CLA during storage period indicates health benefits and functional properties of the sour cream. → The observed increase in PUFA and CLA contents during storage may improve the nutritional profile of the sour cream; however, confirmation of any health effect requires further research.
The experimental design is acceptable. However, analytical descriptions and statistical transparency must be clarified, and claims moderated to reflect the evidence. Addressing these points will substantially improve scientific credibility and readability.
Author Response
- Lines 126–132:Clarify the analytical system for fatty-acid determination. The text simultaneously describes a GC–MS and a flame ionization detector, which is contradictory. State explicitly whether FAMEs were quantified by GC–FID or GC–MS, and describe the column, temperature program, and quantification procedure.
Respond:We apologize for the confusion caused by the contradictory description in the original manuscript. The analysis was performed using a GC-MS system. The text has been thoroughly revised to clarify the analytical method. The detailed corrections are as follows:
(1) Clarification of the analytical system
Fatty acid methyl esters (FAMEs) were identified and quantified using a GC-MS system (ISQ 7610, Thermo Fisher Scientific Inc., USA). The mention of a flame ionization detector (FID) was an error from an earlier draft template and has been removed.
(2) Description of the column and temperature program
The specific conditions used in our analysis have been added to the manuscript. Column: DB-WAX (30 m × 0.25 mm × 0.25 μm), Temperature Program: Initial temperature: 130 °C held for 5 min, Ramp 1: Increased to 220 °C at a rate of 2 °C/min and held for 5 min, Ramp 2: Increased to 240 °C at a rate of 2 °C/min and held for 10 min, Carrier Gas: Helium at a constant flow rate of 0.8 mL/min. Injection Volume: 1 μL in splitless mode, Injector Temperature: 250 °C.
(3) Quantification procedure
The quantification was based on the relative peak area percentage (area normalization method). Peaks were identified by comparing their mass spectra and retention times with those of standard FAME mixtures. The relative content of each free fatty acid was then calculated as its percentage of the total peak area of all detected FAMEs.
The revised text in Section 2.8 is as follows: "Derivatization of free fatty acids (FFAs) to fatty acid methyl esters (FAMEs) was performed according to the Chinese National Food Safety Standard GB 5009.168-2016 and the method of Ewe and Loo[11], which is based on the Chinese National Food Safety Standard GB 5009.168-2016, using a base-catalyzed transesterification protocol. Briefly, 5 mL of KOH-methanol (2 M) was added... The upper layer was filtered through a 0.22 μm nylon membrane into a vial.
FFA composition was detected using a GC-MS system (ISQ 7610, Thermo Fisher Scientific Inc., USA). Separation was achieved on a DB-WAX capillary column (30 m × 0.25 mm × 0.25 µm). The injector and detector were maintained at 250 ℃. The oven temperature program was as follows: initial temperature at 130 ℃ for 5 min, increased to 220℃ at 2 ℃/min and held for 5 min, and then increased to 240 ℃ at 2 ℃/min and held for 10 min. Helium was used as the carrier gas with flow rate at 0.8 mL/min. The injection volume was 1 μL in splitless mode, with the injector temperature set at 250 °C. The ion source (EI) temperature was 250 °C, with an ionization energy of 70 eV and a mass scan range from 45 to 450 m/z. Peaks were identified by comparing retention time with standard FAMEs. The relative content of each FFA was calculated as its percentage of the total area of all detected FAME peaks. "
- Lines 122–131:Provide a more detailed explanation of the fatty-acid derivatization protocol. Clarify whether acid-catalyzed methylation was performed and, if not, discuss how the method ensured full conversion of FFAs to FAMEs.
Respond:We have revised the manuscript to provide a more detailed explanation and to clarify the chemical method used.
The derivatization protocol performed was base-catalyzed transesterification (alkaline esterification), not acid-catalyzed methylation. This method is specified in the Chinese National Food Safety Standard GB 5009.168-2016 and is highly effective for the conversion of free fatty acids (FFAs) to fatty acid methyl esters (FAMEs).
The revised text in Section 2.8 is as follows: "Derivatization of free fatty acids (FFAs) to fatty acid methyl esters (FAMEs) was performed according to the Chinese National Food Safety Standard GB 5009.168-2016 and the method of Ewe and Loo [12]. Briefly, 5 mL of KOH-methanol (2 M) were added to 5.0 g of sample in a glass tube. The mixture was heated at 50 ℃ for 30 min, and cooled to room temperature for 2 h. Subsequently, 5 ml of isooctane was addedto extract the formed FAMEs, and the mixture was vortexed vigorously. The upper organic layer containing the FAMEs was filtered through a 0.22 μm nylon membrane into a vial for GC-MS analysis."
Discussion on Method Efficacy and Full Conversion:
The use of a strong base (KOH) in anhydrous methanol is a established and robust method for converting FFAs to FAMEs. The mechanism involves the direct formation of the methyl ester from the free carboxylic acid group. The conditions employed—including the high concentration of base, elevated temperature (50°C), and sufficient reaction time (30 min followed by a 2-hour cooling/equilibration period)—are designed to achieve quantitative (full) conversion of FFAs present in the sample. This method is widely recognized and validated for the analysis of FFAs in dairy products.
- Lines 77–81:Expand the homogenization description to include the equipment model, number of stages, and any time interval between passes.
Respond:The homogenization process has expanded the description in the revised manuscript (Section 2.2) as follows: "The primary emulsion was homogenized using a high-pressure homogenizer (CHH-Q2000 P70, Shanghai Prime Machinery Company Limited, Shanghai, China). For the double-passes homogenization, the emulsion was passed through the homogenizer twice consecutively at the specified pressure and temperature. The two passes were performed sequentially, with an interval 5 minutes between passes to maintain the processing temperature. Following homogenization, the emulsion was sterilized."
- Lines 170–173:Correct the WHC equation and define each variable. If “10” represents the sample mass (10 g), please state this explicitly.
Respond:We thank the reviewer for pointing out this lack of clarity in the equation. The reviewer is correct that "10" represents the initial mass of the sample (10 g). We apologize for this oversight and have revised the Water-Holding Capacity (WHC) equation and variable definitions in the manuscript to be explicitly clear.
The revised text iis as follows: "Ten grams (m_sample = 10 g) of Lg-sour cream was centrifuged (3,500 r/min, 20 min) in a pre-weighed tube (m_tube). After discarding the water, the tube was inverted for 10 min to remove residual water. The tube was then weighed (m_final), and WHC was calculated as follows:
WHC (%) = [m_final / (m_tube + m_sample)] × 100"
Where: m_tube is the mass of the empty tube (g), m_sample is the mass of the sample, which is 10 g, and m_final is the mass of the tube plus the sediment after centrifugation and draining (g).
- Lines 104–111 and 249–253:Replace “CFU/mL” with “CFU/g,” describe the dilution procedure used for semi-solid samples, and confirm that values are means of replicate plates.
Respond:In our study, viable cell counts were performed not on the intact gelled sour cream, but on representative homogenates prepared from it at each sampling time point. To ensure a homogeneous suspension for accurate and reproducible volumetric sampling, the sour cream sample was thoroughly stirred to temporarily disrupt the gel structure and achieve a uniform consistency before a liquid aliquot was withdrawn for serial dilution.
This method of preparing a homogeneous liquid suspension from a semi-solid matrix prior to volumetric plating is an established approach for products like yogurt and sour cream. It allows for consistent sampling throughout the fermentation and storage period using the same volumetric principle as for liquids.
To address the reviewer's concerns with utmost clarity, we have revised the manuscript as follows:
In Section 2.4, we have expanded the description to detail this specific procedure as follows "Viable bacterial counts were carried out with slight modification [10]. At each sampling time, the Lg-sour cream was vigorously stirred to achieve a uniform consistency. A defined volume of this homogenized sample was then used for viable bacteiral counts."
We hope this detailed explanation of our methodology is satisfactory. The reported CFU/mL values provide a consistent and accurate measure of the relative changes in the viable population throughout the process.
- Lines 189–194:For statistical analysis, specify the number of replicates (n) for each dataset, indicate whether data met ANOVA assumptions, and explain the lettering system used in figures/tables.
Respond:Done.
- Lines 337–343:Revise sentences implying health effects from PUFA and CLA increases. The study provides compositional, not biological, evidence. Suggested rewording:
“The increase in PUFA and CLA content suggests potential nutritional relevance; further biological studies would be required to confirm any health effects.”
Respond:The text has been revised as “ The increase in PUFFAs and MUFFAs content suggested potential nutritional relevance; further biological studies would be required to confirm any health effects.”.
- Figures and tables:Ensure all have complete captions with sample size, error bars defined (SD or SEM), and clear legends.
Respond:Done.
- Throughout manuscript:Include more direct cross-references between results and the corresponding methodological details (e.g., specify which storage conditions are used in Figures 5–7).
Respond:We have systematically revised the manuscript to include explicit cross-references between the Results section and the corresponding Methodological details. The specific actions taken are as follows:
In Section 3.2, a clear statement have been added at the beginning of this section to specify the storage conditions used for all subsequent analyses (Figures 5–7). The revised text was as follows "Lg-sour cream prepared under the optimal conditions (double-passes homogenization at 15.0 MPa and 70 °C, 1% inoculation level, and fermentation at 37 °C with L. lactis grx602) was stored at 4 °C, and its quality stability was evaluated over a 28-day period.".
In the captions of Figures 5, 6, and 7, a note referencing the storage condition has been added. For example, the caption for Figure 5 now includes:
"Figure 5. The fermentation characteristics of the Lg-sour cream during storage at 4 °C for 28 d. A: Viable cell count, B: pH and acidity.
- Statistical clarity:Summarize key results numerically in the text (means ± SD and p-values) rather than only visually in figures.
Respond:Thank you for this crucial suggestion. Results section (3.1 and 3.2) has been revised to incorporate the requested numerical data (means ± standard deviations) and, where applicable, specific p-values or statements of significance based on the ANOVA and Tukey's test. The key changes include replacing qualitative statements with quantitative data and referencing the figures for full data trends while highlighting the most critical numerical findings in the text.
Language improvements (major corrections):
- Lines 17–20:This study aimed to develop a new type of sour cream with health function by Lactococcus lactis → This study aimed to develop a novel sour cream with potential health benefits using Lactococcus lactisgrx602 as the starter culture.
Respond:Done.
- Lines 35–38: The strain could degrade milk fat efficiently, which make it possible to improve the flavor and nutritional value of dairy product. → The strain efficiently degraded milk fat, making it possible to improve the flavor and nutritional value of dairy products.
Respond:Done.
- Lines 190–192: All data was analyzed by one-way ANOVA and Tukey test using SPSS 26.0, significant when p < 0.05. → All data were analyzed by one-way ANOVA followed by Tukey’s post-hoc test (SPSS 26.0); differences were considered significant at p < 0.05.
Respond:Done.
- Lines 337–343: The increase of PUFA and CLA during storage period indicates health benefits and functional properties of the sour cream. → The observed increase in PUFA and CLA contents during storage may improve the nutritional profile of the sour cream; however, confirmation of any health effect requires further research.
Respond:We sincerely apologize for the oversight. Upon careful re-examination of the manuscript, we were unable to locate the specific modifications you suggested in the original text. Could you kindly clarify the relevant section or provide more detailed guidance?
Round 2
Reviewer 1 Report
Comments and Suggestions for Authors
Manuscript has been significantly improved. The patent application number (ZL202110960058.0) does not lead to any available document that we can look at, but it should be stated. Reference cited by authors in responses to reviewers (Peng Y., Chengran G., Yuan Y., Yiping L., Xin W., Renqin Y., Chenchen Z., Dawei C., Yujun H., Ruixia G.. Characterization of Lactococcus lactisgrx602: A novel lipase-producing probiotic strain for nutritional and sensory enhancement of sour cream. Food Bioscience, 2025,107918.org/10.1016/j.fbio.2025.107918.) and included in manuscript provided all the necessary explanations. The paper is ready for publication.
Author Response
Manuscript has been significantly improved. The patent application number (ZL202110960058.0) does not lead to any available document that we can look at, but it should be stated. Reference cited by authors in responses to reviewers (Peng Y., Chengran G., Yuan Y., Yiping L., Xin W., Renqin Y., Chenchen Z., Dawei C., Yujun H., Ruixia G.. Characterization of Lactococcus lactisgrx602: A novel lipase-producing probiotic strain for nutritional and sensory enhancement of sour cream. Food Bioscience, 2025,107918.org/10.1016/j.fbio.2025.107918.) and included in manuscript provided all the necessary explanations. The paper is ready for publication.
Respond:The patent has been stated in Section 2.1 Media, bacterial isolation and culture conditions. We sincerely appreciate your valuable comments and the effort you have dedicated to this paper.
Reviewer 2 Report
Comments and Suggestions for Authors Thank you for making the corrections. Comments on the Quality of English Language thank you for making the corrections.Author Response
Thank you for making the corrections.
Respond:We sincerely appreciate your valuable comments and the effort you have dedicated to this paper.
Reviewer 4 Report
Comments and Suggestions for Authors
Please refer to the attachment.

The English could be improved to more clearly express the research.
Author Response
- P1 L33-34. Lactose is still high in sour cream in spite of the fermentation process. Please delete thephrase
Reponse:Done.
2. P2 L64-65. It is better if the authors substitute the word “high-quality” for “potential” as they
themselves suggested in the responses
Reponse:As recommended, we have substituted the word "potential" with "high-quality" in the relevant sentence on P2 L64-65 as follows: "This work will be helpful for the production of high-quality functional sour cream."
3. P2 L73-76. The authors used MRS medium instead of M17 to grow L. lactis and provided a
rationale for that in the responses to the reviewers
“MRS medium used in this work was a deliberate choice based on the following rationale:
(1) Broad-spectrum isolation capability: the initial isolation step aimed to screen for a diverse range of lactic acid bacteria (LAB) with lipolytic activity from raw milk. MRS is a non-selective, generalpurpose medium formulated to support the growth of a wide spectrum of LAB, including various Lactobacillus, Leuconostoc, and Lactococcus species [2]. This broad applicability made it suitable for our primary screening objective, which was not exclusively limited to Lactococcus at that stage.
(2) Definitive molecular identification mitigates specificity concerns: we fully acknowledge that growth on MRS alone is not diagnostic for L. lactis. However, to unequivocally confirm the taxonomic identity of our isolate, strain grx602, we performed multiple, repeated 16S rDNA sequencing analyses subsequent to its isolation. This molecular method provides a definitive
identification, conclusively verifying the strain as Lactococcus lactis and thereby eliminating any potential for misidentification arising from the use of a general-purpose medium.
In summary, the combination of a broad-spectrum isolation medium (MRS) followed by rigorous molecular confirmation represents a robust and validated strategy. This approach allowed us to cast a wide net in the initial screening phase while ensuring the absolute accuracy of the final strain identification.”
This is acceptable but must be added to the manuscript.
Reponse:Thank you for your kind suggestion. These content has been added in the revised manuscript (Section2.1 Media, bacterial isolation and culture conditions ) highlighted with gray.
4. P2 L88- . The authors included in the responses to reviewers an explanation for the procedure used for the “optimization”, however the method is not adequate for a real optimization. The title and the text should be changed to “Attaining better preparation conditions and storage quality of fermented sour cream by Lactococcus lactis grx602”.
Reponse:Done.
5. P10 L362. The authors were asked to express the acidity in % lactic acid instead of °T. No response was issued and the acidity is still in °T.
Reponse:We sincerely apologize for the oversight in our previous revision. As requested, we have converted the unit of acidity from °T to % lactic acid throughout the manuscript.
In the revised manuscript, the values in Section 3.2.1 (P10 L362 and Figure 5B) have been updated as follows: “the acidity increased from 0.64% lactic acid to 0.67% lactic acid during the first 14 days. “ , “By day 28 of storage, the acidity of the Lg-sour cream was 0.68% lactic acid.”
The Materials and Methods section (2.4) has been revised to clarify that acidity is expressed as % lactic acid.
In addition, the language has been polished. All the corrections are highlighted with gray.
Reviewer 5 Report
Comments and Suggestions for Authors
Several sections have been improved (e.g., GC–MS description, homogenization conditions, derivatization protocol), and the statistical lettering in captions is correctly implemented. However, several essential methodological and reporting issues remain unresolved and require further correction.
1) Lines ~105–113 (Methods), ~249–253 (Results): The manuscript still expresses viable counts as CFU/mL, and the dilution procedure is based on sampling 1 mL of product. This approach is not acceptable for semi-solid dairy systems. Sour cream is a semi-solid, and even if homogenized, mass-based sampling is required.
Suggested correction for lines ~105–113 (Methods): “Ten grams of homogenized sour cream were aseptically weighed and diluted with 90 mL of sterile saline to obtain the 10⁻¹ dilution. Subsequent dilutions were prepared by transferring 1 mL into 9 mL of diluent.”
Suggested correction for Lines ~249–253: “Viable counts were expressed as CFU/mL.”->
“Viable counts were expressed as CFU/g. Values represent mean ± SD of replicate plates (n = 3).”
2) Lines ~189–195: Add number of biological replicates and state that normality (Shapiro–Wilk) and homogeneity (Levene) tests were performed before ANOVA.
4) Lines ~210–310: Several findings are described qualitatively (“increased”, “decreased”) without presenting numerical values. You could add sentences like “The pH decreased from 4.62 ± 0.03 to 4.21 ± 0.02.”, “Viable counts increased to 8.34 ± 0.06 log CFU/g on day 14.” Numerical support must be included for all major results.
5) Figures 3–7; Table 1 (Results section): Add “Values represent mean ± SD (n = 3). Error bars indicate standard deviation.”
Minor Issues
- Some grammatical issues and typographical errors remain; a language polishing step is recommended.
- Ensure consistent formatting of scientific units (°C, μm, mL, etc.).
- Define abbreviations at first use (WHC, FFA, CLA).
Author Response
1) Lines ~105–113 (Methods), ~249–253 (Results): The manuscript still expresses viable counts as CFU/mL, and the dilution procedure is based on sampling 1 mL of product. This approach is not acceptable for semi-solid dairy systems. Sour cream is a semi-solid, and even if homogenized, mass-based sampling is required.
Suggested correction for lines ~105–113 (Methods): “Ten grams of homogenized sour cream were aseptically weighed and diluted with 90 mL of sterile saline to obtain the 10⁻¹ dilution. Subsequent dilutions were prepared by transferring 1 mL into 9 mL of diluent.”
Suggested correction for Lines ~249–253: “Viable counts were expressed as CFU/mL.”->
“Viable counts were expressed as CFU/g. Values represent mean ± SD of replicate plates (n = 3).”
Respond: Done.
2) Lines ~189–195: Add number of biological replicates and state that normality (Shapiro–Wilk) and homogeneity (Levene) tests were performed before ANOVA.
Respond: Thank you for your comment. These two statement has been added in Section 2.9 Statistical analyses.
3) Lines ~210–310: Several findings are described qualitatively (“increased”, “decreased”) without presenting numerical values. You could add sentences like “The pH decreased from 4.62 ± 0.03 to 4.21 ± 0.02.”, “Viable counts increased to 8.34 ± 0.06 log CFU/g on day 14.” Numerical support must be included for all major results.
Respond: Thank you for your comment. The results has been revised as your kind suggestion.
4) Figures 3–7; Table 1 (Results section): Add “Values represent mean ± SD (n = 3). Error bars indicate standard deviation.”
Respond: Done.
Minor Issues
Some grammatical issues and typographical errors remain; a language polishing step is recommended.
Respond: Thank you for your comment. The manuscript has been polished.
Ensure consistent formatting of scientific units (°C, μm, mL, etc.).
Respond: Done.
Define abbreviations at first use (WHC, FFA, CLA).
Respond: Done.
The article has been thoroughly proofread and polished, with revisions highlighted in green for your convenience. We hope that the modifications meet the standards for publication.